# Monkeys mutant for *PKD1* recapitulate human autosomal dominant polycystic kidney disease

Tomoyuki Tsukiyama [1,2,10]*, Kenichi Kobayashi[1,3,10], Masataka Nakaya[1,2,10], Chizuru Iwatani[1], Yasunari Seita[1], Hideaki Tsuchiya[1], Jun Matsushita[1], Kahoru Kitajima[1], Ikuo Kawamoto[1], Takahiro Nakagawa[1], Koji Fukuda[4], Teppei Iwakiri[4], Hiroyuki Izumi[4], Iori Itagaki[1,5], Shinji Kume[6], Hiroshi Maegawa[6], Ryuichi Nishinakamura[7], Saori Nishio[8], Shinichiro Nakamura[1], Akihiro Kawauchi[3] & Masatsugu Ema [1,2,9]*

Autosomal dominant polycystic kidney disease (ADPKD) caused by *PKD1* mutations is one of the most common hereditary disorders. However, the key pathological processes underlying cyst development and exacerbation in pre-symptomatic stages remain unknown, because rodent models do not recapitulate critical disease phenotypes, including disease onset in heterozygotes. Here, using CRISPR/Cas9, we generate ADPKD models with *PKD1* mutations in cynomolgus monkeys. As in humans and mice, near-complete *PKD1* depletion induces severe cyst formation mainly in collecting ducts. Importantly, unlike in mice, *PKD1* hetero-zygote monkeys exhibit cyst formation perinatally in distal tubules, possibly reflecting the initial pathology in humans. Many monkeys in these models survive after cyst formation, and cysts progress with age. Furthermore, we succeed in generating selective heterozygous mutations using allele-specific targeting. We propose that our models elucidate the onset and progression of ADPKD, which will serve as a critical basis for establishing new therapeutic strategies, including drug treatments.

[1] Department of Stem Cells and Human Disease Models, Research Center for Animal Life Science, Shiga University of Medical Science, Shiga 520-2192, Japan. [2] Institute for the Advanced Study of Human Biology (WPI-ASHBi), Kyoto University, Kyoto 606-8501, Japan. [3] Department of Urology, Shiga University of Medical Science, Shiga 520-2192, Japan. [4] Shin Nippon Biomedical Laboratories, Ltd, Kagoshima 891-1394, Japan. [5] The Corporation for Production and Research of Laboratory Primates, Ibaraki 305-0003, Japan. [6] Department of Medicine, Shiga University of Medical Science, Shiga 520-2192, Japan. [7] Department of Kidney Development, Institute of Molecular Embryology and Genetics, Kumamoto University, Kumamoto 860-0811, Japan. [8] Division of Rheumatology, Endocrinology and Nephrology, Hokkaido University Graduate School of Medicine, Hokkaido 060-8648, Japan. [9] PRESTO, Japan Science and Technology Agency, Saitama 332-0012, Japan. [10]These authors contributed equally: Tomoyuki Tsukiyama, Kenichi Kobayashi, Masataka Nakaya. *email: ttsuki@belle.shiga-med.ac.jp; mema@belle.shiga-med.ac.jp

Autosomal dominant polycystic kidney disease (ADPKD) is one of the most common hereditary diseases, with an incidence at birth estimated to be 1 in 400 to 1 in 1000 (refs. [1–3]), and is 10, 15, and 20 times more common than sickle cell disease, cystic fibrosis, and Huntington's disease, respectively[4]. It is estimated that there are six million patients with ADPKD worldwide, more than half of whom develop end-stage renal disease and require dialysis or kidney transplantation by age 60 (ref. [5]). Although numerous important findings have been reported and many treatments have been proposed for ADPKD based on studies using animal disease models and tissue culture models[6–11], and in fact some drugs have been used to slow symptom progression[12], no definitive therapies are currently available.

It is reasonable to assume that physiological and genetic differences explain why medications for ADPKD yield different responses between small animals and humans. The pathological recapitulation of human diseases is limited in mouse models because there are marked differences in physiology between humans and mice. Some diseases, such as Parkinson's disease, Alzheimer's disease, AIDS, and influenza, cannot be recapitulated in mice[13]. Most human ADPKD patients are *PKD1* heterozygotes with a germline mutation in one allele[14,15]. Manifestations of the disease generally do not appear for decades, but a large-scale study by Reed et al.[16] showed that renal cysts developed in 46% of 420 children with a family history of ADPKD. Given that half of children harbor the causative mutation, 92% of ADPKD children should eventually demonstrate cyst development. Early cyst formation was also observed in heterozygous pigs with induced mutations and in cats and dogs with spontaneous mutations[17–19]. By contrast, heterozygous deletion of *Pkd1* in mice rarely results in the formation of cysts until near the end of life, around 1.5 years after birth[20,21]. Although there are many types of mouse ADPKD models, including *Pkd1* or *Pkd2* knockout (KO), conditional KO, and induction of hypomorphic mutations, consistent cyst formation requires biallelic mutations in most models[22]. Additionally, other disease model animals used in polycystic kidney disease studies, such as jck mice, pcy mice, Han:SPRD-Cy rats, and PCK rats, have mutations in other genes such as *Nek8*, *Nphp3*, *Pkdr1*, and *Pkhd1*, respectively. Thus, rodent models do not precisely recapitulate the human disease state. To overcome these limitations, we aimed to create a novel ADPKD model animal in a species closely related to humans, the cynomolgus monkey. Among species that have been generated via genome editing with CRISPR/Cas9, the cynomolgus monkey is physiologically and genetically one of the closest species to humans[23].

Here, we generate *PKD1* KO or mosaic monkeys with various degrees of cyst formation by using CRISPR/Cas9 technology, and reveal the lineages of cyst epithelial cells. Furthermore, we generate heterozygous monkeys with conventional or allele-specific gene targeting, which reveal the presence of cysts in the perinatal stage, representing the earliest manifestations of the disease.

## Results

**PKD1 mutant monkeys with varying degrees of cystic severity.** For efficient introduction of loss-of-function mutations, we designed sgRNAs within exon 2 of *PKD1* because frameshift mutations in this region can result in the loss of a large percentage of polycystin-1 (PC1), the translation product of *PKD1* (ref. [24]) (Fig. 1a). We then selected the most efficient sgRNA using a single-strand annealing (SSA) assay[25] (Supplementary Fig. 1a, b).

To determine the optimal experimental parameters, we injected one of three concentrations (50, 100, or 200 ng/μl) of Cas9 mRNA and 50 ng/μl of sgRNA into cynomolgus embryos. Overall, 207 of

423 embryos that survived after injection developed into blastocysts. Among the 207 embryos, 71 were used for genotyping, 86 were used for embryo transfer to generate monkeys, and the remaining 50 were frozen (Fig. 1b). A T7 endonuclease I (T7E1) assay of blastocyst embryos collected at day 8 showed that almost all embryos had mutations in the targeted region. The percentage with mutations was 88.7% overall (63/71 embryos) (Fig. 1b, Supplementary Fig. 1c–e). DNA sequencing confirmed that the embryos had indel mutations near the Cas9 cleavage site (Supplementary Figs. 1f–2b). This efficiency was comparable to or even higher than that in a previous study[23]. Despite such high efficiency, the blastocyst development rate of the *PKD1*-mutated embryos (46.7%) was comparable to that of wild-type (WT) embryos (43.8%)[26], although some developmental retardation was observed (Supplementary Fig. 2c). A comparison of mutation rates in each embryo at the three Cas9 mRNA concentrations showed that the use of 200 ng/μl Cas9 mRNAs tended to induce higher mutation rates than 50 and 100 ng/μl (Fig. 1c). However, we decided to use 50 or 100 ng/μl Cas9 mRNAs to increase the production of live animals, considering the embryonic lethality observed in *Pkd1*-deficient mice[21].

Among 86 embryos that were transferred individually to surrogate mothers (Fig. 1d, ET), 29 were implanted. The pregnancy rate (33.7%) of mothers with implanted embryos was comparable to that of mothers with WT embryos (8/23, 34.8%) that were transferred by single-embryo transfer at our institute from 2016 to 2018. Subsequently, there were 7 miscarriages before embryonic day 150 and 8 stillbirths after day 151, and 14 fetuses were delivered. Most of the aborted monkeys died in the perinatal stage, because the gestational period in cynomolgus monkeys is approximately 160 days (Fig. 1e, day of death). It was difficult to pinpoint the cause of death in aborted monkeys, but at least in some monkeys, placental abruption was considered to be primarily responsible. Placental abruption is also often seen in WT monkeys. The live birth rate of the *PKD1*-mutated fetuses (14/29 fetuses, 48.3%) was lower than that of WT fetuses (7/8 fetuses, 87.5%), indicating developmental defects in the *PKD1*-mutated fetuses during pregnancy. Four neonates died spontaneously after birth; thus, 10 live monkeys were obtained (Fig. 1d, e). In total, kidney samples were recovered from 12 of 19 aborted or dead monkeys (Fig. 1e). After genotyping by DNA sequencing, these aborted or dead animals were divided into KO, mosaic, and heterozygous groups (Fig. 1e, Supplementary Fig. 3). Additionally, kidneys of the animals in the KO or mosaic group were classified by histological analysis into three categories: (1) severe type, (2) intermediate type, and (3) mild type (Fig. 1e, f, Supplementary Fig. 3). In some monkeys (Mild #1, Severe #1, Severe #3), many tissues (kidney, liver, spleen, stomach, pancreas, small intestine, colon, uterus, bladder, ovary, adrenal glands, thymus, thyroid, heart, lung, brain, skeletal muscle, skin, placenta, amnion, and umbilical cord) were used for genotyping. However, we could not detect clear variations in genotypes among tissues. Given that mutations were induced early in development, it was appropriate to observe mosaicism within individual tissues, not between tissues. Therefore, in other monkeys, genomic DNA samples were collected from the kidneys of dead animals, and from the placentas, amnions, and umbilical cords of living monkeys to reduce invasiveness. Severe-type kidneys showed formation of numerous cysts, encompassing more than 30% of the specimen area, and similar morphology to end-stage ADPKD kidneys in human heterozygotes and a human infant with biallelic *PKD1* abnormalities[27]. Intermediate-type kidneys demonstrated moderate cyst formation, covering 5–30% of the specimen area. Mild-type kidneys exhibited sporadic cyst formation, encompassing less than 5% of the specimen area, and similar morphology

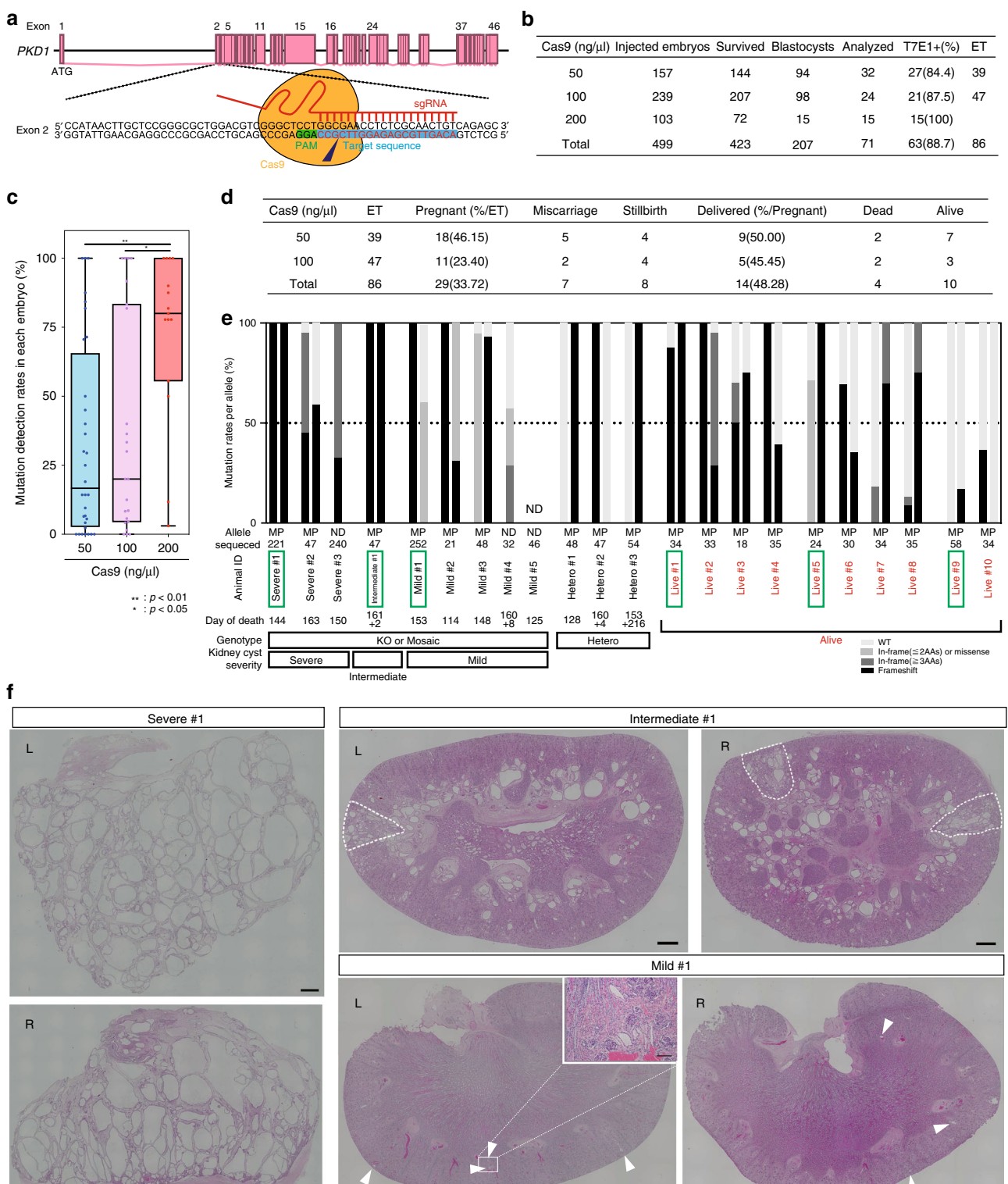

** : p < 0.01
* : p < 0.05

to early-stage ADPKD kidneys in human heterozygotes[28,29] (Fig. 1f, Supplementary Fig. 3). In monkeys with intermediate-type kidneys, it is noteworthy that there were several kidney compartments in which more than 30% of the area was filled with multiple cysts (Fig. 1f, the compartments surrounded by dotted lines), similar to monkeys with severe-type kidneys, while cysts in other compartments developed sporadically (Fig. 1f). This suggests that intermediate-type kidneys may constitute a transitional state between severe- and mild-type kidneys, since it had features of both. Collectively, although the phenotype is variable,

due to mosaicism, the severity of the phenotype tended to correlate with the mutation rate.

**PKD1-KO monkeys show enlarged kidneys with numerous cysts.** Three of 15 aborted fetuses showed extreme abdominal distension. The kidneys were greatly enlarged (Fig. 2a, Supplementary Fig. 4a) and innumerable cysts were observed by ultrasonography and on cross-sections of bisected kidneys (Fig. 2b, c, Supplementary Fig. 4b). Additionally, the fetuses had immature lungs, indicating the presence of Potter's phenotype with death

**Fig. 1** *PKD1* mutant monkeys with varying degrees of cystic severity. **a** A schematic diagram of the targeting site in the monkey *PKD1* gene. **b** The numbers of embryos injected with 50, 100, or 200 ng/μl Cas9 mRNAs and 50 ng/μl sgRNA. "Survived" indicates the number of survived embryos after injection. We used embryos developed to blastocysts for genotyping analysis (Analyzed) or embryo transfer (ET) and the remaining embryos were frozen. **c** Box plot of mutation detection rates in each mRNA-injected embryo. The top and bottom edges of boxes indicate the first and third quartiles, respectively; the center lines indicate the medians, and the ends of whiskers indicate the maximum and minimum values, respectively. $n = 71$ biologically independent samples. **d** The numbers of aborted fetuses and delivered monkeys. **e** Categorization of genotypes of aborted embryos and delivered monkeys. Black bars indicate the rates of frameshift mutations. Dark gray bars indicate the rates of in-frame mutations involving more than two amino acids. Light gray bars indicate the rates of one- or two-amino-acid in-frame mutations and missense mutations. Off-white bars indicate wild-type (WT) rates. In the row labeled "Allele," "M" indicates the maternal allele, "P" indicates the paternal allele, and "ND" indicates that it was undetermined whether the alleles were maternal or paternal. "Sequenced" indicates the number of sequences examined in this analysis. "Day of death" indicates the day of abortion. In cases of monkeys that died after birth, the number before "+" indicates the delivery day and the number after "+" indicates the day of death. Green boxes indicate the animals whose data are shown in **f** and Fig. 5a. **f** Low-power, H&E-stained images indicating severe-, intermediate-, and mild-type kidneys. The compartments in intermediate-type kidneys that show features similar to those in severe-type kidneys are surrounded by dotted lines. "L" indicates left kidneys and "R" indicates right kidneys. Arrowheads indicate cyst formation. Scale bar in large images, 1 mm. Scale bar in the small box, 100 μm. Source data are provided as a Source Data file.

from pulmonary hypoplasia (Fig. 2d, Supplementary Fig. 4c). Histological analysis of the kidneys showed interstitial fibrosis and cysts with thin epithelial linings (Fig. 2e). This phenotype is similar not only to that of *Pkd1* null mice[21] but also to that of a human infant with biallelic *PKD1* abnormalities[27]. Confirming the genotyping results (Supplementary Fig. 3a), immunofluorescence analysis with an anti-PC1 antibody detected virtually no signals[30], indicating near-complete depletion of the PC1 protein (Fig. 2f). Western blot analysis also showed that the severe-type kidney did not express full-length PC1 protein (Supplementary Fig. 4d). Immunohistochemistry analysis using anti-Ki67 revealed proliferation of several epithelial cells in a few small cysts, but there was no clear signal in the majority of expanded cysts (Supplementary Fig. 4e). Furthermore, several cysts were observed in monkeys' liver and pancreatic ducts (Fig. 2g, h, Supplementary Fig. 4f, g), confirming that as in *Pkd1* KO mice[20,21], PC1 plays an important role in these organs. Taken together, depletion of *PKD1* in cynomolgus monkeys resulted in a similar phenotype to that seen in *PKD1*-null humans and mice[21,27].

**Mimicking the result of several "two-hit" mutations**. In human ADPKD patients with heterozygous mutations in one allele, cysts may be formed after many decades when a second mutation occurs in another allele[2]. Although this "two-hit" hypothesis is still controversial, we reasoned that our mosaic animals may at least partially mimic this situation. In contrast to the near-complete depletion of *PKD1*, several cysts developed sporadically in monkeys with mild-type kidneys (Fig. 3a). Although the majority of normal-appearing renal tubules expressed PC1 at similar levels as in a WT monkey, cells lining the cysts did not express PC1 (Fig. 3a, b). It was unclear whether the absence of PC1-staining in cysts could be attributed to lack of PC1 or to expansion of cysts derived from segments with lower signal intensity (i.e., those from WT cells). However, most of the WT cortex was PC1 positive, and no extensive PC1-negative areas were observed; in contrast, in mosaic kidneys there were extensive PC1-negative areas. Therefore, we judged that the extensive PC1-negative areas were those where biallelic defects occurred as a result of genome editing. Interestingly, a fraction of normal-appearing renal tubules around the PC1-negative cysts also hardly expressed PC1 (Fig. 3a, arrowheads), suggesting that intercellular communication between WT and *PKD1*-null cells may ameliorate the cyst severity of mutant cells, or that the loss of PC1 protein is not sufficient for cyst formation and some other factor may be required, as stipulated by the "three-hit" hypothesis[31].

**The lineage identities of cyst epithelial cells**. To investigate the origin of the cysts, the sections were stained with lotus tetragonolobus lectin (LTL), a marker for the proximal renal tubule; aquaporin-1 (AQP1), a marker for the proximal renal tubule and the descending limb of Henle's loop; uromodulin (UMOD), a marker for the thick ascending limb of Henle's loop; Na–Cl cotransporter (NCC), a marker for the distal convoluted tubule; aquaporin-2 (AQP2), a marker for the connecting tubule and the collecting duct; and E-cadherin (ECAD), a marker for the region from the ascending limb of Henle's loop to the collecting ducts (Fig. 4a). An examination for autofluorescence or non-specific staining using negative control immunoglobulin resulted in no clear signal. Additionally, we were able to separately stain each nephron segment using AQP1, AQP2, NCC, and UMOD antibodies (Supplementary Fig. 5a). In severe-type kidneys, most cysts were AQP2 positive, while a few cysts were positive for AQP1, NCC, or UMOD (Fig. 4b, Supplementary Fig. 5b), indicating that most cysts were derived from the connecting tubule or the collecting ducts. Importantly, the AQP2-positive cysts were larger than the others (Fig. 4c, Supplementary Fig. 5c). Consistent with this observation, it was reported that the ratio of AQP2- to AQP1-positive cysts increased with cyst enlargement in human end-stage ADPKD kidneys[28], and similar results have been reported in mouse models of ADPKD[32,33].

In intermediate-type kidneys, AQP1-positive cysts developed from various structures (Fig. 4d, Supplementary Fig. 6a). The sizes of AQP1-positive cysts in the outer medulla were larger than those of cysts in other areas (Fig. 4e). However, in compartments where multiple cysts comprised more than 30% of the area (Fig. 1f, the compartments surrounded by dotted lines), most of these cysts were AQP2 positive, similar to those in severe-type kidneys (Fig. 4f, Supplementary Fig. 6b), suggesting that cyst formation in AQP2-positive segments is more severe than in other segments.

In mild-type kidneys, the majority of cysts were AQP1 positive and developed in the outer medulla, consistent with intermediate-type kidneys (Fig. 4g, Supplementary Fig. 6c–e), whereas several AQP2-positive cysts developed in the cortex and were larger in size (Fig. 4g, h). Interestingly, the sizes of AQP1-positive cysts in mild- and intermediate-type kidneys were comparable to those in severe-type kidneys, whereas AQP2-positive cysts in severe-type kidneys were much larger than those in mild- and intermediate-type kidneys (Supplementary Fig. 6f). It was reported that cysts derived from the proximal tubules were smaller than those from the collecting ducts in humans with end-stage ADPKD and in several mouse models, and that the majority of cysts in the early stage were derived from the proximal segment in a progressive mouse model[28,32,33]. Thus, these results suggest that

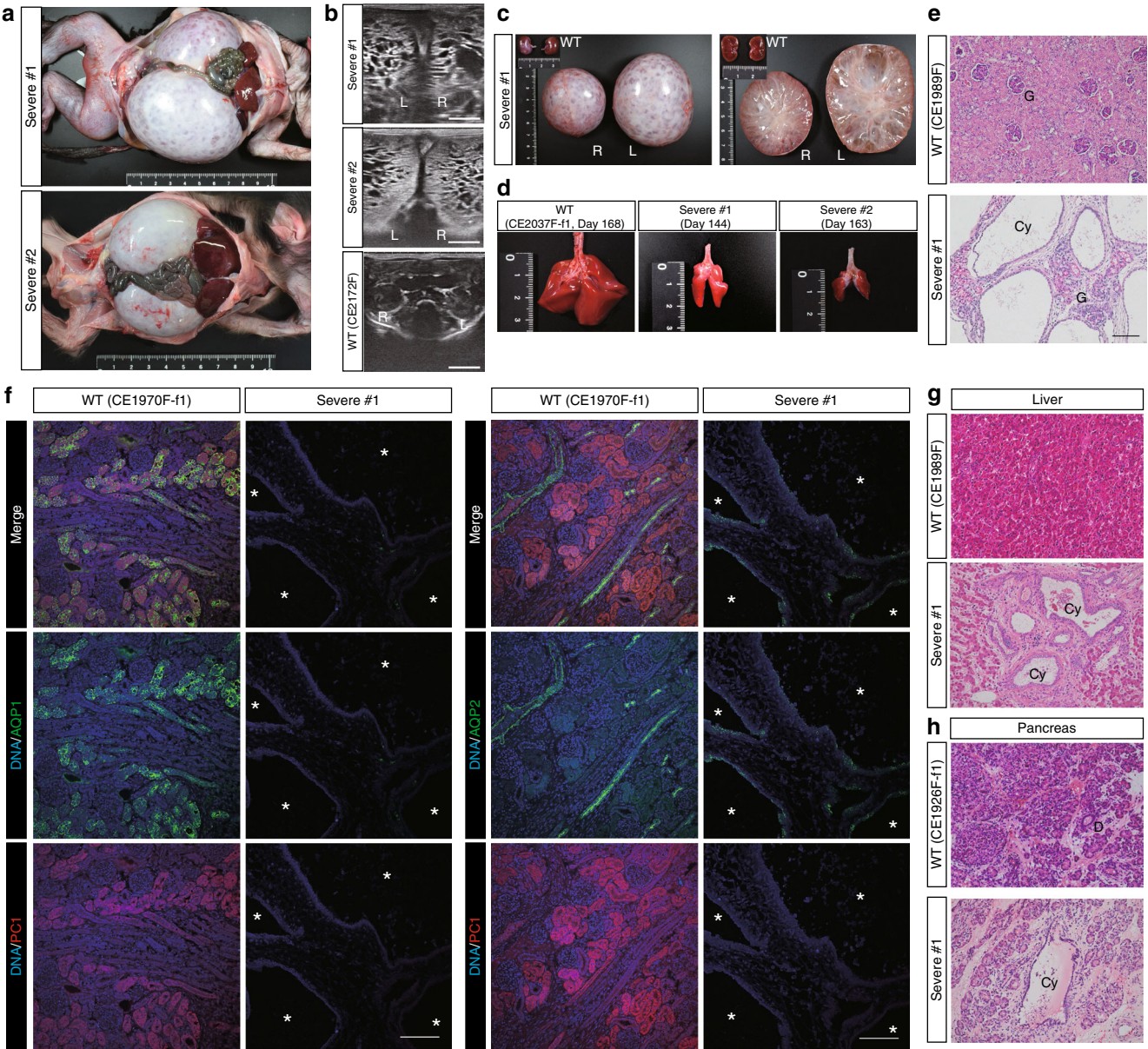

**Fig. 2** *PKD1*-KO monkeys show enlarged kidneys with numerous cysts. **a** Enlarged kidneys in monkeys with severe-type kidneys. **b** Ultrasonography of the kidneys. "L" indicates left kidneys and "R" indicates right kidneys. **c** The gross appearances and cross-sections of severe-type kidneys and WT kidneys on the same scale. **d** The immature lungs of monkeys with severe-type kidneys and a WT monkey on the same scale. "Day" indicates the day of abortion. **e** H&E-stained image of the kidneys. "G" indicates glomeruli and "Cy" indicates cysts. Scale bar, 100 μm. **f** The expressions of AQP1, AQP2, and PC1 in a severe-type kidney. Asterisks indicate cysts. Scale bar, 100 μm. **g** Liver cysts. "Cy" indicates cysts. Scale bar, 100 μm. **h** A pancreatic cyst. "Cy" indicates a cyst and "D" indicates a duct. Scale bar, 100 μm.

AQP1-positive, proximal tubule-derived cysts may form prior to AQP2-positive, collecting duct-derived cysts.

Furthermore, ECAD-positive and AQP1-, AQP2-, NCC-, and UMOD-negative cysts were observed in all kidney types (Fig. 4i), suggesting that these cysts were common in *PKD1*-mutant monkeys. Because NCC is a marker for one portion of the distal tubule, these cysts may have been derived from other segments of the distal tubule. Collectively, cysts originated from several nephron epithelial types, including AQP1-positive proximal tubules, ECAD-positive distal tubules, and AQP2-positive collecting ducts, in both monkeys with near-complete deletion of *PKD1* and mosaic monkeys, while cyst formation in the collecting ducts was associated with cyst severity because extensive or high-density cyst formations were detected in AQP2-positive collecting ducts.

**PKD1-mutated monkeys with numerous cysts can survive**. Ten live monkeys were followed up by ultrasonography of kidneys and blood biochemical examination. Although no detectable cysts were observed until 9 months after birth in the kidneys of Live #9, which had a low mutation rate, several cysts developed at 6 months after birth in the kidneys of Live #5, which had a mutation rate of more than 50% (Fig. 5a, Supplementary Fig. 3f). Furthermore, 3 months after that, one new cyst developed in the left kidney and one cyst in the right kidney enlarged (Fig. 5a, Live #5). Interestingly, numerous cysts were already detected prior to 2 months after birth in the kidneys of Live #1, which had a high mutation rate (Fig. 5a, Supplementary Fig. 3f). Two monkeys with 93.8% and 69.6% frameshift mutations (Live #1 and #4, respectively) showed formation of numerous cysts. On the other hand, two monkeys with 8.6% and 18.2% frameshift mutations (Live #9

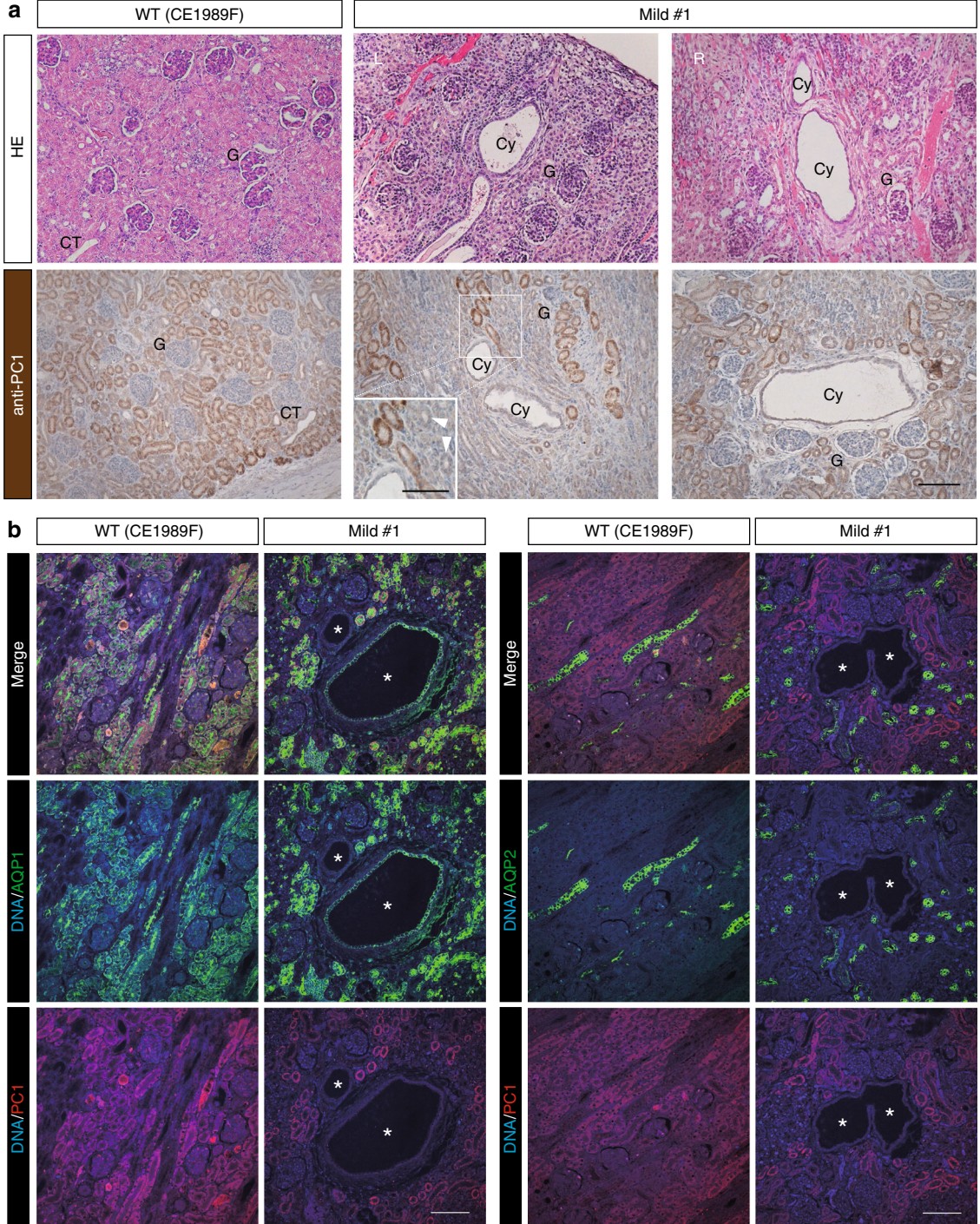

**Fig. 3** Mimicking the result of several "two-hit" mutations. **a** H&E-stained images and the expressions of PC1 in mild-type kidneys. The upper row of H&E-stained images shows cyst formation in the kidneys. The lower row of immunohistochemically stained images shows the expression of PC1. "G" indicates glomeruli, "Cy" indicates cysts, and "CT" indicates collecting ducts. Arrowheads indicate PC1-negative tubules. Scale bar, 100 μm. **b** The expressions of AQP1, AQP2, and PC1 in a mild-type kidney. Asterisks indicate cysts. Scale bar, 100 μm.

and #10, respectively) showed no cyst formation. The other six monkeys showed mild cyst formation (Supplementary Table 1), indicating that the frequency of cyst formation tended to correlate with the mutation rate of the *PKD1* gene.

Blood tests at 6 and 12 months after birth showed no remarkable abnormalities in any monkeys demonstrating cyst formation (Fig. 5b), indicating that nephron compensation occurred in monkeys as it does in humans. This result is reasonable because it is known that in humans with progressive renal enlargement, nephron compensation enables the glomerular filtration rate to remain within the normal range for several decades, until more than 50% of the functioning parenchyma has been destroyed[5]. Given that monkeys have longer life spans than rodents, and the cyst formation in our primate models resembled not only that in late-stage human patients but also that in pediatric patients, we can monitor the disease state over the long term and thereby elucidate the molecular mechanisms of ADPKD and assessing drug efficacy and toxicity.

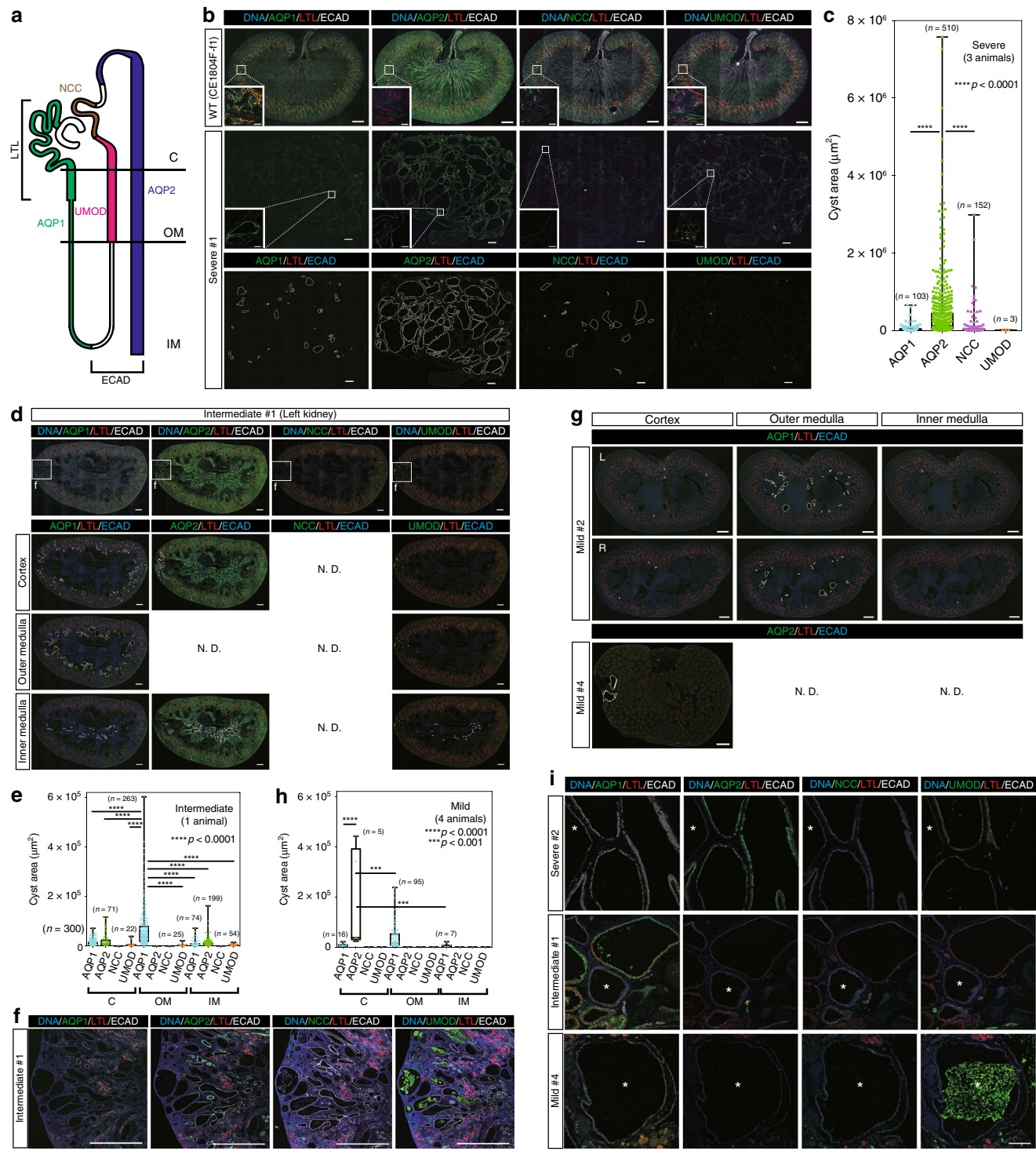

**Heterozygotes exhibit distal-tubules-derived small cysts.** Heterozygous animals had the WT sequence in one allele and a mutation in the other, which is likely also the case in human ADPKD patients (Supplementary Fig. 7a). Kidneys in heterozygotes showed formation of a few cysts perinatally (Fig. 6a, Supplementary Fig. 7b). Although most of the cysts were small, we confirmed that these cysts were rarely detected in WT monkeys aborted around full-term gestation (Supplementary Fig. 7c). Notably, a clear cyst was detected by ultrasonography in a live heterozygous monkey (Hetero #3), and this cyst was confirmed on the cut surface of the kidney (Fig. 6a). These results clearly

demonstrated the presence of cysts perinatally in heterozygous monkeys, which is similar to the situation in heterozygous humans but not mice[16,20,34].

While small numbers of cysts in the heterozygotes were AQP2 positive, most were ECAD positive but negative for the other markers (Fig. 6b, c, Supplementary Fig. 7d). This type of cyst also formed in monkeys with near-complete deletion of *PKD1* and in mosaics, as described above (Fig. 4i). Thus, this cyst type may precede the formation of AQP1- or AQP2-positive cysts and may represent the earliest manifestation of the disease, although long-term monitoring is required for definite proof. Most cysts in

**Fig. 4** The lineage identities of cyst epithelial cells. **a** Schematic diagram showing nephron segments positive for each marker. "C" indicates the cortex, "OM" indicates the outer medulla, and "IM" indicates the inner medulla. **b** Expressions of nephron segment markers in a severe-type kidney. Small boxes indicate regions containing representative cysts positive for AQP1, AQP2, NCC, or UMOD, respectively; cyst areas that are positive for each marker are surrounded by white lines. Scale bar in large images, 1 mm. Scale bar in small boxes, 100 μm. **c** Box plot of the areas of AQP1-, AQP2-, NCC-, or UMOD-positive cysts in three severe-type kidneys. $n = 768$ cysts in three animals. **d** Expressions of nephron segment markers in an intermediate-type kidney. Small boxes indicate the regions shown at high magnification in **f**. Cyst areas that are positive for AQP1, AQP2, or UMOD, respectively, are surrounded by white lines. ND indicates not detected. Scale bar, 1 mm. **e** Box plot of the areas of AQP1-, AQP2-, NCC-, or UMOD-positive cysts in the intermediate-type kidneys. "C" indicates the cortex, "OM" indicates the outer medulla, "IM" indicates the inner medulla. $n = 1008$ cysts in one animal. **f** Expressions of nephron segment markers in an intermediate-type kidney compartment containing multiple cysts similar to those in severe-type kidneys. Scale bar, 1 mm. **g** Expressions of nephron segment markers in mild-type kidneys. The cyst areas that are positive for AQP1 or AQP2 in the cortex, the outer medulla, or inner medulla are surrounded by white lines. "L" indicates a left kidney and "R" indicates a right kidney. ND indicates not detected. Scale bar, 1 mm. **h** Box plot of the areas of AQP1-, AQP2-, NCC-, or UMOD-positive cysts in intermediate-type kidneys. "C" indicates the cortex, "OM" indicates the outer medulla, and "IM" indicates the inner medulla. $n = 123$ cysts in four animals. **i** Representative ECAD-positive and AQP1-, AQP2-, NCC-, and UMOD-negative cysts. Asterisks indicate the cysts. Scale bar, 100 μm. In the box plots, the top and bottom edges of boxes indicate the first and third quartiles, respectively; the center lines indicate the medians; and the ends of whiskers indicate the maximum and minimum values, respectively. Source data are provided as a Source Data file.

heterozygotes were PC1 negative, consistent with the "two-hit" hypothesis (Fig. 6d, Hetero #1). The formation of multiple PC1-negative cysts in early age is surprising because it is thought that multiple cyst formation results from the accumulation of second mutations with age. Interestingly, several PC1-positive cysts were also observed (Fig. 6d, Hetero #3). We examined the presence of autofluorescence or non-specific staining using negative control immunoglobulin at the same concentration as anti-PC1 antibody (20 μg/ml) and detected no clear signal (Supplementary Fig. 7e). This suggests that these cysts may be caused by *PKD1* haploinsufficiency, although there is a possibility that the signal was non-specific because some extra bands were detected in western blotting (Supplementary Fig. 4d). Consistent with this observation, it was reported that some cysts were PC1 positive in humans[35,36]. These results are in sharp contrast to those in mice, where efficient cyst formation was shown to require biallelic mutations[22].

In humans, therapies for ADPKD are limited; thus, active therapeutic intervention is not recommended in pediatric patients with no remarkable symptoms[37]. Consequently, in ADPKD patients the pathological process from the presymptomatic stage to the early stage remains a black box. Understanding how cysts develop and worsen in the early stage in heterozygotes may be useful in designing drugs to prevent cysts from enlarging.

**Generation of heterozygotes by allele-specific targeting.** Using the conventional method to target exon 2, we were able to obtain five heterozygotes, including two animals from which kidney samples were not collected, but the efficiency was low (5 of 29 monkeys; Supplementary Table 1). Since Ma et al.[38] succeeded in correcting a specific pathogenic allele in human embryos at high efficiency, we attempted to target a single allele using allele-specific polymorphisms. We searched for polymorphisms in the monkey *PKD1*-coding region and found that a region of exon 4 had a single-nucleotide polymorphism. Oocytes and sperm from monkeys bred in different countries (China and Indonesia) were used to generate embryos with this polymorphism in the maternal allele. In these embryos, a guide RNA (gRNA) targeting the region was unable to recognize the maternal allele, but could specifically recognize the paternal allele; thus, mutations were induced only in the paternal allele (Fig. 7a). To reduce the off-target effect in this experiment, instead of injecting mRNA 6 h after intracytoplasmic sperm injection (ICSI), we co-injected a gRNA/Cas9 protein RNP complex with sperm or electroporated the RNP complex into embryos immediately after ICSI[38,39]. In an in vitro experiment using a low concentration (20 ng/μl) of Cas9 protein, incomplete mutagenesis and mosaicism were observed

(Supplementary Fig. 8a). Therefore, we decided to use a high concentration (200 ng/μl) of Cas9 protein to completely introduce mutations into single-cell embryos and thus avoid mosaicism.

Two fetuses (Ex4 Severe #1 and Ex4 Hetero #1) were aborted spontaneously and two live monkeys (Ex4 Live #1 and Ex4 Live #2) were obtained. One aborted fetus (Ex4 Severe #1) had frameshift mutations in both alleles and showed a similar phenotype to monkeys with *PKD1* depletion (Supplementary Fig. 8b–g), clearly demonstrating that the phenotype was caused by mutations in *PKD1* and not by gRNA off-target effects, because two different gRNAs were able to induce a similar phenotype. DNA sequencing showed that the other three monkeys were heterozygotes (Supplementary Fig. 8h, i). To date, eight monkeys have additionally been generated and 10 of 12 monkeys have been shown to be heterozygotes or monoallelically affected. Moreover, among these 10 monkeys, nine have a single mutation, and therefore only 3 of 12 monkeys demonstrate mosaicism. This is quite different from exon 2 targeting, where most animals exhibit mosaicism, illustrating a dramatic improvement in our ability to generate non-mosaic heterozygotes (Supplementary Table 1). Histological analysis detected small cysts in the kidneys of the aborted fetus (Ex4 Hetero #1) (Fig. 7b), and the cysts were ECAD positive and AQP1, AQP2, NCC, and UMOD negative (Fig. 7c, d). These results were similar to those in exon 2 heterozygotes, indicating the shared biological characteristics of exon 2 and exon 4 heterozygotes. Overall, most of the cysts in the heterozygotes were ECAD positive, while some cysts were also positive for AQP2 and were larger than the others (Fig. 7e). Furthermore, the kidneys in the two living monkeys (Ex4 Live #1 and Ex4 Live #2) were followed up by ultrasonography, and formation of a few cysts was detected at birth or by 2 months after birth, respectively (Fig. 7f, g). These cysts had enlarged at 6 months after birth (Fig. 7g, right images). Because some kidney cysts were identified in human heterozygous children[16,34], we propose that the heterozygous monkeys in this study may serve as a useful model for the study of human ADPKD, including pediatric patients.

**Generation of a floxed allele in monkey embryos.** Finally, we applied conditional KO to *PKD1* mutagenesis to verify in detail the mechanism of cyst formation according to the "two-hit" hypothesis. We used a long single-strand oligonucleotide (long ssODN)[40–43] with two substituted bases and three gRNAs to efficiently create a floxed/indel monkey at the F0 generation (Fig. 8a). Three of 30 embryos had a floxed allele, as determined by restriction fragment length polymorphism (RFLP) analysis (Fig. 8b, c) and DNA sequencing (Fig. 8d). Genomic fragments

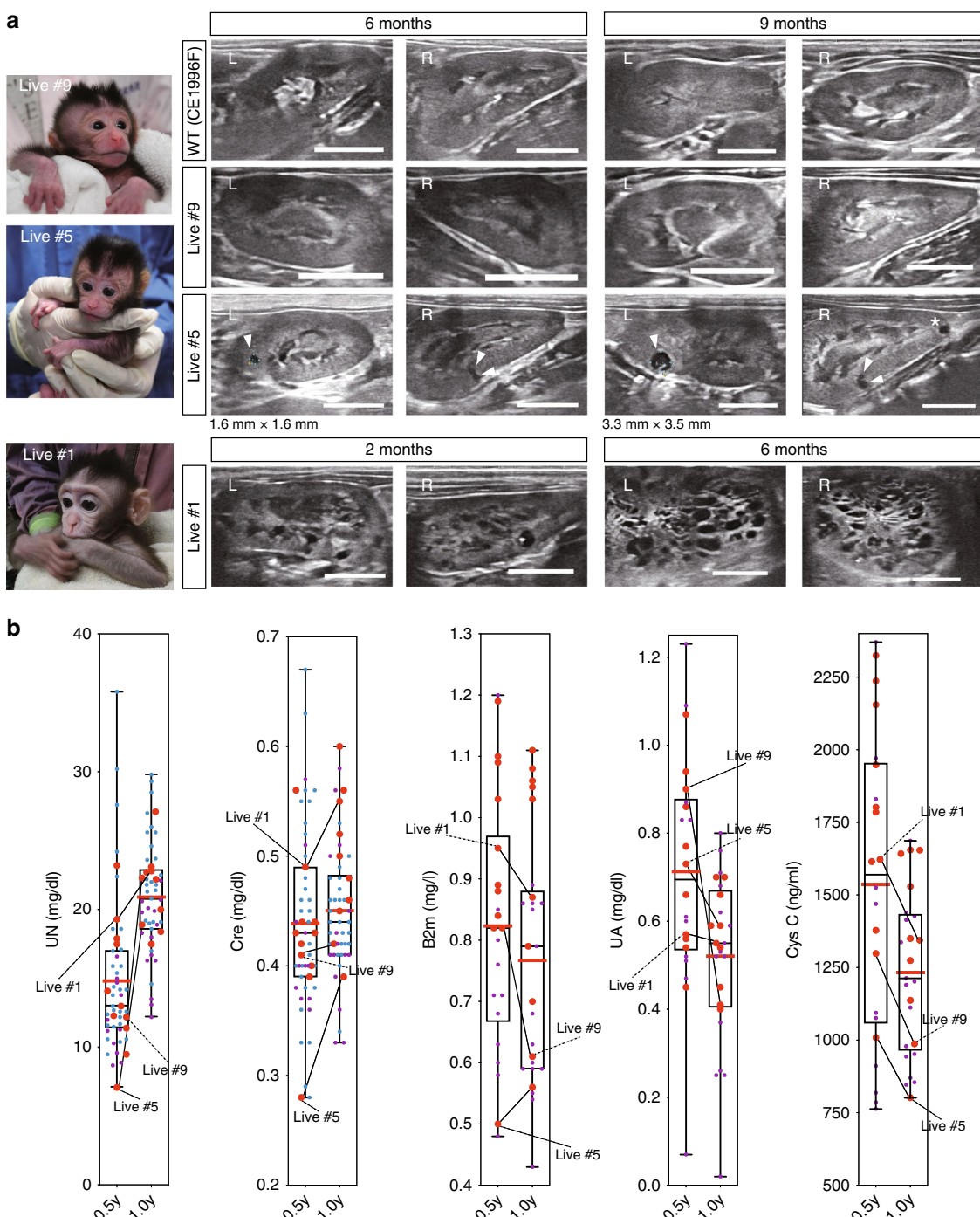

**Fig. 5** *PKD1*-mutated monkeys with numerous cysts can survive. **a** Cyst formation detected by ultrasonography in live monkeys. Arrowheads indicate cyst formation. Scale bar, 10 mm. **b** Box plot of renal function of each monkey at 6 and 12 months after birth. "UN" indicates urea nitrogen, "Cre" indicates creatinine, "B2m" indicates beta2-microglobulin, "UA" indicates uric acid, and "Cys C" indicates cystatin C. Large red dots indicate mutated monkeys. Small purple dots indicate wild-type monkeys grown at Shiga University of Medical Science. Small blue dots indicate wild-type monkeys grown at Shin Nippon Biomedical Laboratories. The top and bottom edges of boxes indicate the first and third quartiles, respectively; the center lines indicate the medians; and the ends of whiskers indicate the maximum and minimum values, respectively. Red lines indicate the average. Source data are provided as a Source Data file. *n* = 55 animals.

containing the outer regions of homologous arms were amplified by PCR (Fig. 8a, blue triangles), and the fragments were cloned into vectors and sequenced. The sequencing analysis confirmed *cis* insertion of LoxP sites in all three floxed embryos (Fig. 8d). In one embryo (Experiment 1, #4), we detected a single LoxP insertion. Additionally, we confirmed the sequence fidelity of junctional regions in two of three floxed embryos. This is the

first successful generation of a floxed allele in nonhuman primates. Introduction of Cre recombinase into this floxed allele would further accelerate the study of ADPKD in primates. Once the floxed monkeys are generated, deletion of *PKD1* in renal tubule cells can be achieved by viral delivery of Cre enzyme. Although viral delivery may cause mosaic Cre expression, we can take advantage of the mosaicism for long-term studies.

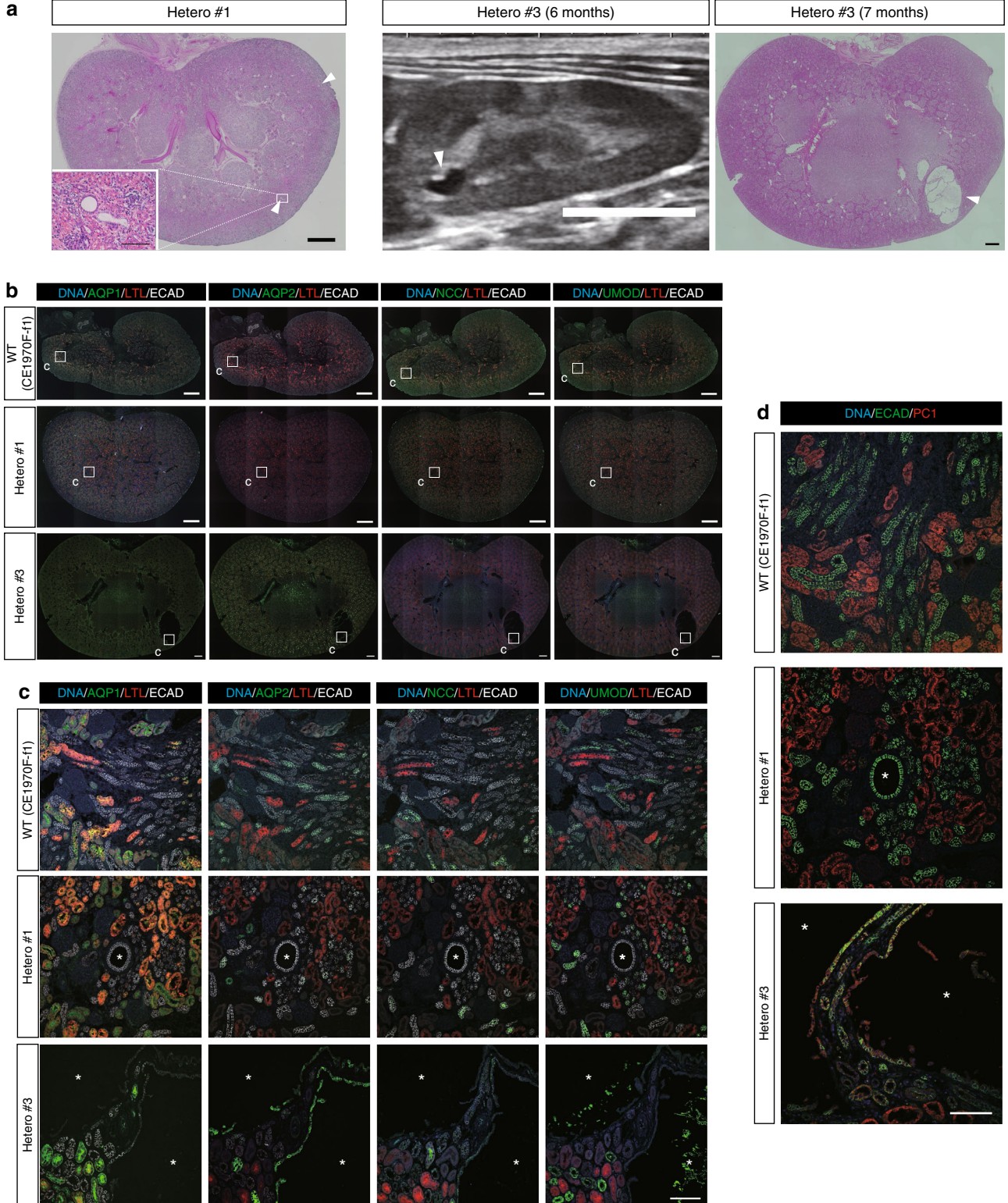

**Fig. 6** Heterozygotes exhibit distal-tubules-derived small cysts. **a** Low-power, H&E-stained images and ultrasonography image of heterozygous kidneys. Scale bars in large H&E-stained images, 1 mm. Scale bar in the small box, 100 μm. Scale bar in the ultrasonography image, 10 mm. **b** Expressions of nephron segment markers in heterozygous kidneys. Small boxes indicate the regions shown at high magnification in **c**. Scale bar, 1 mm. **c** Representative ECAD-positive and AQP1-, AQP2-, NCC-, and UMOD-negative cysts and AQP2-positive cysts in heterozygous kidneys. Asterisks indicate cysts. Scale bar, 100 μm. **d** Expressions of ECAD and PC1 in heterozygous kidneys. One representative PC1-negative cyst in Hetero #1 and two PC1-positive cysts in Hetero #3 are shown. Asterisks indicate cysts. Scale bar, 100 μm.

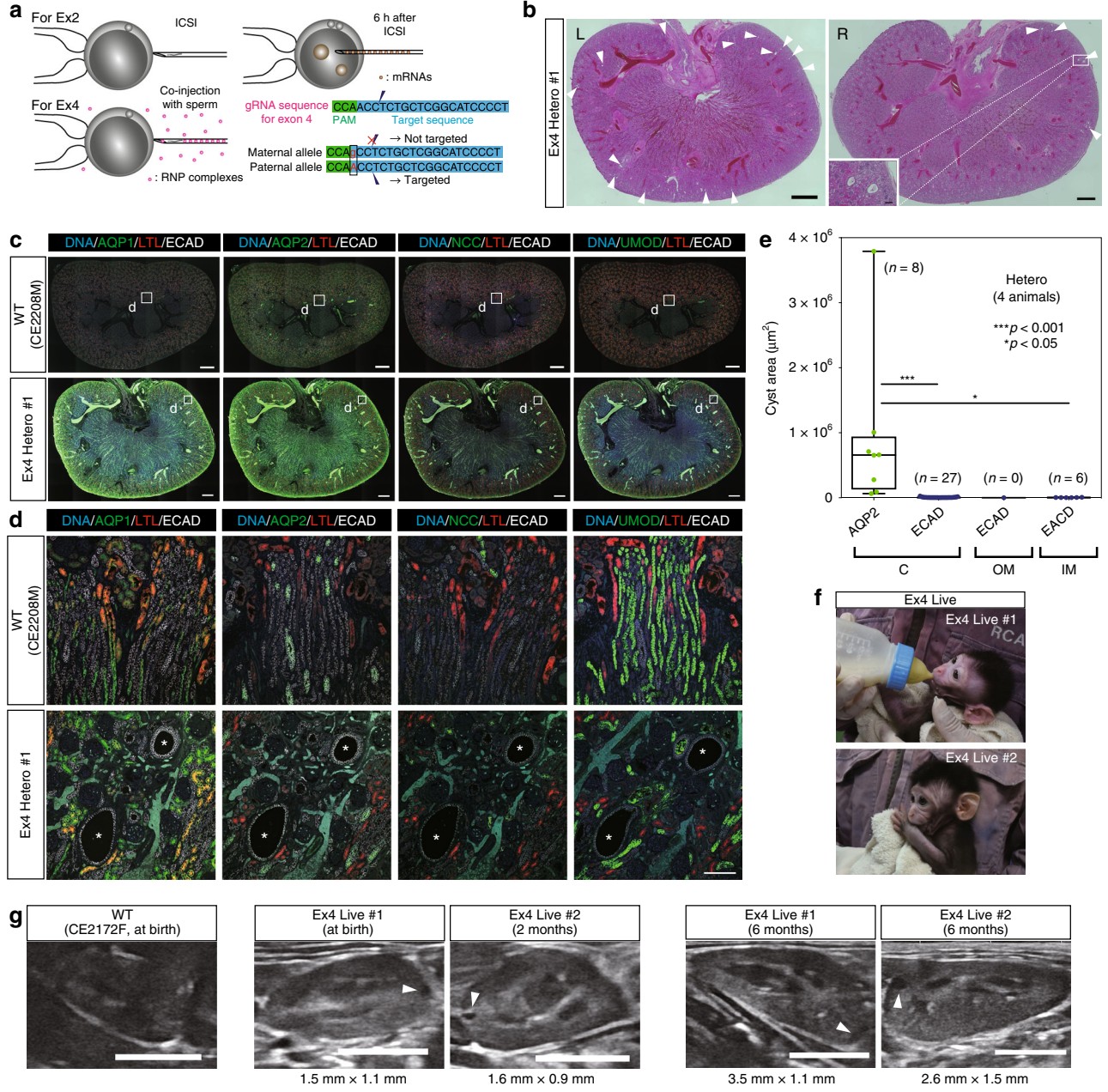

**Fig. 7** Generation of heterozygotes by allele-specific targeting. **a** Schematic diagram showing the difference between exon 2 targeting and exon 4 targeting. In exon 2 targeting, mRNAs were injected into zygotes 6 h after ICSI. In exon 4 targeting, gRNA/HiFi Cas9 protein RNPs were co-injected with sperm into MII oocytes. Polymorphism prevents excision of the gRNA target sequence in the maternal allele. **b** Low-power, H&E-stained image of exon 4 heterozygous kidneys. "L" indicates a left kidney and "R" indicates a right kidney. Arrowheads indicate cyst formation. Scale bars in large images, 1 mm. Scale bar in the small box, 100 μm. **c** Expressions of nephron segment markers in an exon 4 heterozygous kidney. Small boxes indicate the regions shown at high magnification in **d**. Scale bar, 1 mm. **d** Representative ECAD-positive and AQP1-, AQP2-, NCC-, and UMOD-negative cysts in exon 4 heterozygous kidneys. Asterisks indicate cysts. Scale bar, 100 μm. **e** Box plot of the areas of ECAD-positive or AQP2-positive cysts in all heterozygous kidneys. The top and bottom edges of boxes indicate the first and third quartiles, respectively; the center lines indicate the medians; and the ends of whiskers indicate the maximum and minimum values, respectively. n = 41 cysts in four animals. **f** Two live monkeys with exon 4 heterozygous mutations. **g** Cyst formation detected by ultrasonography in exon 4 heterozygotes. Arrowheads indicate cyst formation. Scale bar, 10 mm. Source data are provided as a Source Data file.

## Discussion

In summary, we generated multiple types of *PKD1* mutants in cynomolgus monkeys. Complete or mosaic deletion led to cyst formation with various degrees of severity, which tended to correlate with the mutation rate. In heterozygotes, it takes a long time for cysts to occur frequently, and therefore it will presumably take time to start preclinical testing. On the other hand, in a mosaic animal with partial biallelic mutation, animals showing some degree of cyst formation can probably be obtained at an early stage and should therefore be useful for preclinical studies. Other large animal models, such as those involving induced mutations in pigs and spontaneous mutations in cats and

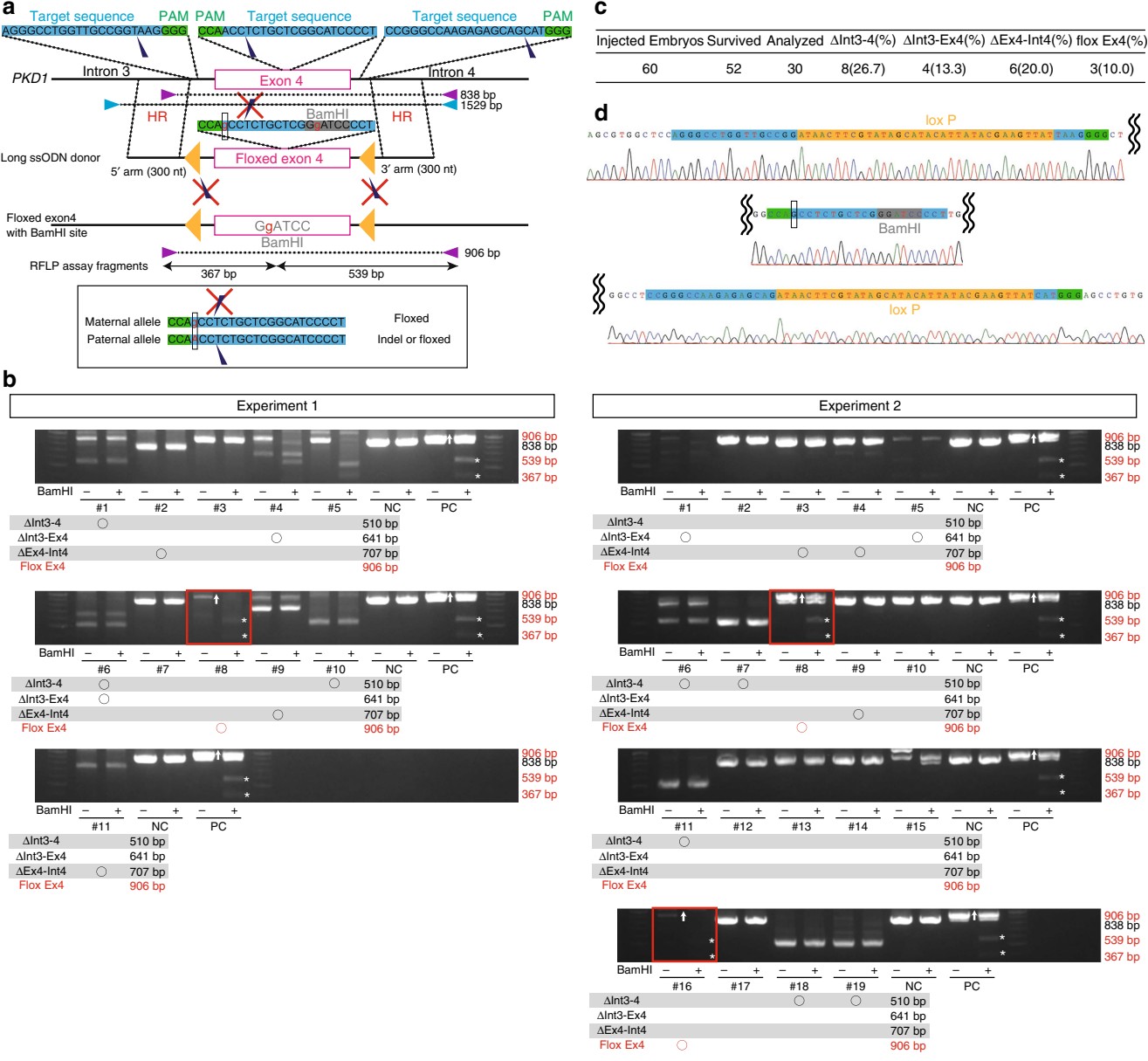

**Fig. 8** Generation of a floxed allele in monkey embryos. **a** Schematic diagram showing the floxed allele knock-in strategy. Purple triangles indicate PCR primers for RFLP assay. Blue triangles indicate PCR primers for sequencing. Yellow triangles indicate loxP sequences. **b** RFLP assay using *Bam*HI for detection of the floxed allele. Asterisks indicate positive bands. "NC" indicates negative control, in which wild-type DNAs were used as PCR templates. "PC" indicates positive control, in which knock-in DNAs were used as PCR templates. $n = 30$ biologically independent samples. **c** Efficiencies of deletion and floxed allele knock-in. **d** The inserted sequence in the floxed allele. Source data are provided as a Source Data file.

dogs, are also useful for preclinical testing. However, monkeys are more similar to humans genetically and physiologically than these animals. Additionally, like in humans, genetic diversity is higher in monkeys than in other model animals. These features are advantageous for preclinical studies. Certainly, it is true that the use of monkeys would involve expensive and long-term studies. However, although ADPKD is a very common monogenic disease that occurs very frequently, it is also true that treatment methods are still limited despite several decades of research thus far. Even if it takes time and money, we believe it is time to try a new research approach.

Heterozygous monkeys have an advantage over mouse models in terms of determining how cysts develop and aggravate in human heterozygotes with ADPKD. In contrast to mice, which require conditional knockout to mimic the human phenotype, it

is feasible to investigate the process of cyst formation in a more natural state using our monkey model. We engineered a *PKD1* allele using allele-specific targeting, and for the first time selectively generated *PKD1* heterozygote nonhuman primates that formed cysts in a similar way as human pediatric patients. Since heterozygotes without mosaicism can be prepared, we believe that they can be studied genetically in a relatively homogeneous population. Although this will of course take time, in the future we can perform this research under more genetically homogeneous conditions in family pedigrees by analyzing the F1 generation. We have not yet evaluated the fertility of the monkeys we generated, but given that human ADPKD patients are fertile, we believe that these monkeys will be fertile and can generate multiple pedigrees. The pathology in heterozygotes was similar to that in humans, so the same pathological progression can be

expected in the F1 generation. Therefore, we hope that elucidating the mechanisms by which cysts develop and worsen may identify treatment that can begin in childhood. Generally, active therapeutic interventions in pediatric patients, including molecular diagnosis, are not recommended, and research in children of ADPKD patients is ethically difficult because ADPKD shows 100% penetrance and there is no definitive therapy. In humans, therefore, it has only been revealed that cysts have developed since childhood, and the details are unknown. Additionally, since rodent models do not precisely recapitulate the disease state in humans, the pathological process from the presymptomatic stage to the early stage in humans remains a black box. We identified the lineage identities of cyst epithelia and found that most cysts in heterozygotes were derived from the distal tubules, which may reflect the initial stage of cystogenesis and represent the earliest manifestations of the disease. Thus, although the collecting ducts are the only drug target in adulthood patients, our results suggest that the distal tubules may be a novel drug target in pediatric patients for whom there is currently no clinical intervention.

We also found that cyst formation in the collecting ducts was associated with cyst severity. Although tolvaptan, a selective vasopressin V2 receptor antagonist, is currently the most common drug used to treat ADPKD patients and effectively suppresses increases in kidney volume[12], the outcomes and side effects after long treatment durations are unclear, especially in pediatric patients. Tolvaptan targets the collecting ducts because they are the location of vasopressin V2 receptor expression. The results in our animal model suggest that severe cysts develop in the collecting ducts, even at a young age. Although initiation of tolvaptan treatment beginning in the early stage of cyst formation is expected to suppress cyst enlargement, it is unfeasible to perform clinical studies in pediatric patients prior to conducting preclinical studies in an animal model. Our monkey model is valuable in this regard as it allows for long-term drug evaluation beginning at a young age.

It is difficult to clearly demonstrate off-target sequence alterations in a genome-wide manner due to limited genomic information in the cynomolgus monkey. Therefore, we used two different gRNAs to show that observed phenotypes were caused by on-target mutations. Interestingly, compared with WT monkeys, we observed higher abortion rates not only with exon 2 targeting but also with exon 4 targeting. In mice, it was shown that loss of Pkd1 caused placental defects[44]. Thus, we suspect that abnormal function of PKD1 may be a cause of the high rate of abortion.

Recently, advances in human kidney organoid research enabled the generation of PKD1-null cyst formation in vitro[45,46]. Although the organoids can directly utilize human cells to unravel the mechanism of cyst formation, these cells are poorly differentiated and the culture period is limited. Some monkeys in our model survive even after the formation of numerous cysts (Fig. 5a). Therefore, we propose that this model is useful for long-term observation, elucidating the molecular mechanisms of ADPKD, and assessing drug efficacy and toxicity.

## Methods

**Study approval.** We followed the Reporting in Vivo Experiments (ARRIVE) guidelines developed by the National Centre for the Replacement, Refinement & Reduction of Animals in Research (NC3Rs). We also followed The Act on Welfare and Management of Animals from Ministry of the Environment, Fundamental Guidelines for Proper Conduct of Animal Experiment and Related Activities in Academic Research Institutions under the jurisdiction of the Ministry of Education, Culture, Sports, Science and Technology, and Guidelines for Proper Conduct of Animal Experiments from Science Council of Japan. All animal experimental procedures were approved by the Animal Care and Use Committee of Shiga University of Medical Science (approval number: 2015-5-13, 2016-6-1).

**Animals.** For oocyte collection, female cynomolgus monkeys (Macaca fascicularis), ranging in age from 4 to 13 years, were selected for this study. In this study, monkeys were never sacrificed and only animals that died naturally were used for sampling. The light cycle was 12 h of artificial light from 8 a.m. to 8 p.m. Each animal was fed 20 g/kg of body weight of commercial pellet monkey chow (CMK-1; CLEA Japan) in the morning, supplemented with 20–50 g of sweet potato in the afternoon. Water was available ad libitum. Temperature and humidity in the animal rooms were maintained at $25 \pm 2\,°C$ and $50 \pm 5\%$, respectively.

**Vector construction.** pX330 was purchased from Addgene (Plasmid #42230)[47]. To construct pX330-monPKD1, oligo-DNAs listed in Supplementary Table 2 were annealed and ligated into the BbsI site of pX330. To construct pCAG-EGxxFP-monPKD1, an amplified PCR product was cloned into the EcoRI–NheI sites of pCAG-EGxxFP[25]. The primers for PCR are listed in Supplementary Table 2.

**SSA assay.** 293FT cells were simultaneously transfected with pX330-monPKD1 and pCAG-EGxxFP-monPKD1[25]. Two days after incubation in Dulbecco's modified Eagle's medium (Sigma) containing 10% fetal bovine serum (Sigma), the cells were collected and analyzed with flow cytometry.

**mRNA in vitro transcription.** To add the T7 promoter sequence to the Cas9 coding region and the sgRNA sequence, PCR amplification was performed[48]. The PCR products were treated with 0.5% SDS, 0.2 mg/ml Proteinase K for 30 min at 50 °C, purified with phenol–chloroform, and precipitated with ethanol. Then, the purified PCR products were used as templates for in vitro transcription. The Cas9 and sgRNA mRNAs were transcribed using the mMESSAGE mMACHINE T7 Transcription Kit and the MEGAshortscript T7 Transcription Kit (Thermo Fisher Scientific), respectively. The mRNAs were purified with the MEGAclear Transcription Clean-Up Kit (Thermo Fisher Scientific). The primers for in vitro transcription are listed in Supplementary Table 2.

**Intracytoplasmic sperm injection.** Two weeks after the subcutaneous injection of 0.9 mg of a gonadotropin-releasing hormone antagonist (Leuplin for Injection Kit; Takeda Chemical Industries), a microinfusion pump (iPRECIO SMP-200, ALZET Osmotic Pumps) with 15 IU/kg human follicle-stimulating hormone (hFSH, Gonapure Injection; Asuka Pharmaceutical) was embedded subcutaneously under anesthesia and injected 7 μl/h for 10 days[49,50]. After the hFSH treatment, 400 IU/kg human chorionic gonadotropin (hCG, Gonatropin; Asuka Pharmaceutical) was injected intramuscularly. Forty hours after the hCG treatment, oocytes were collected by follicular aspiration using a laparoscope (LA-6500, Machida Endoscope). Cumulus-oocyte complexes (COCs) were recovered in alpha modification of Eagle's medium (MP Biomedicals, Solon), containing 10% serum substitute supplement (Irvine Scientific). The COCs were stripped off cumulus cells with 0.5 mg/ml hyaluronidase (Sigma Chemical). ICSI was carried out on metaphase II (MII)-stage oocytes in mTALP containing HEPES with a micromanipulator. Fresh sperm were collected by electric stimulation of the penis with no anesthesia.

**Microinjection of mRNA into ICSI embryos.** For mRNA microinjection, pronuclear-stage embryos were prepared by 6-h culture after ICSI. Following ICSI, embryos were cultured in CMRL Medium-1066 (Invitrogen) supplemented with 20% bovine serum (Invitrogen) at 38 °C in 5% $CO_2$ and 5% $O_2$. Microinjections of mRNA were performed under microscopy with a micromanipulator. A mixture of 50, 100, or 200 ng/μl Cas9 mRNA and 50 ng/μl sgRNA were injected into the cytoplasm of embryos after 6 h of ICSI.

**T7 endonuclease I (T7E1) assay.** To reduce amplification bias, the genomic DNAs from embryos were amplified with the REPLI-g Mini Kit (Qiagen). The targeted sequences were amplified from the samples and purified with Wizard SV Gel and the PCR Clean-Up System (Promega). The purified sequences were denatured, reannealed, and digested with T7E1 (NEB) for 2 h at 37 °C. The electrophoresis was performed using 2% agarose gels. To extract genomic DNAs from tissue samples, the samples were digested in lysis buffer (10 mM Tris-HCl (pH 8.0), 100 mM NaCl, 50 mM EDTA, 0.5% SDS, and 0.5 mg/ml Proteinase K). The lysate was treated with phenol and phenol–chloroform and precipitated with ethanol, followed by denaturing, reannealing, and digesting with T7E1.

**DNA sequencing.** For DNA sequencing, the PCR products with mutations detected by T7E1 assay were cloned into the EcoRV site of pBRBlue II. For each sample, multiple PCR products from different tubes were cloned to reduce bias. To determine the genotypes of fetuses and offspring, genomic DNA samples were collected from the kidneys of dead animals, as well as from the placentas, amnions, and umbilical cords of living monkeys. The parental origin of each sequence was determined by polymorphism analysis.

**Embryo transfer.** When embryos developed to expanded blastocysts, one embryo was transferred into each appropriate recipient female[49,50]. Embryos were aspirated into a catheter (ETC3040SM5-17; Kitazato Medical Service) under a stereomicroscope.

The catheter was inserted into the oviduct of the recipient via the fimbria under the laparoscope, and the cultured embryo was transplanted with a small amount of medium. Pregnancy was determined by ultrasonography 30 days after ICSI.

**Tissue sectioning and hematoxylin–eosin (H&E) staining**. For tissue sectioning, the samples were fixed in Bouin's solution overnight at 4 °C, washed in 70% ethanol, and embedded in paraffin. The paraffin blocks were sectioned at a thickness of 2–4 μm on a microtome and mounted on glass slides (Platinum Pro, Matsunami). The paraffin sections were de-paraffinized with xylene followed by rehydration. Each slide was stained with H&E.

**Immunohistochemistry**. For immunohistochemistry, the paraffin sections were de-paraffinized with xylene followed by rehydration. Each slide was autoclaved for 20 min at 121 °C in Histofine solution (Nichirei Biosciences) for antigen retrieval. The slides were then washed with PBS and incubated in 0.3% $H_2O_2$/methanol for 30 min at room temperature to inactivate endogenous peroxidase. After washing three times with PBS, the slides were blocked with 2% normal goat serum/PBS for 30 min at room temperature and incubated with primary antibodies in the blocking solution for 1 h at room temperature. After washing three times with PBS, the slides were incubated with HRP-conjugated secondary antibodies in the blocking solution for 1 h at room temperature. After washing three times with PBS, the slides were incubated in 0.1 mg/ml DAB/0.03% $H_2O_2$/50 mM Tris-HCl (pH 7.6) for 3–5 min at room temperature. Finally, after hematoxylin counterstaining and hydrophobization, the sections were mounted with mounting medium (MP500, Matsunami).

For immunofluorescence analysis, after antigen retrieval the slides were permeabilized with 0.2% Triton-X/PBS for 30 min at room temperature. After washing with PBS, the slides were blocked with 10% normal donkey serum/2% skim milk/0.1% Tween-20/PBS for 1 h at 4 °C and incubated with primary antibodies in the blocking solution overnight at 4 °C. After washing three times with 0.1% Tween-20/PBS, the slides were incubated with Alexa 488-, 546-, 594-, 633-, or 647-conjugated secondary antibodies and Hoechst 33342 in 2% skim milk/0.1% Tween-20/PBS for 1 h at room temperature or for 4 h at 4 °C. After washing three times with 0.1% Tween-20/PBS, the slides were mounted with mounting medium (Vectashield; Vector Laboratories). The antibodies used in this study were as follows: anti-PC1 (7E12) (sc-130554, 1:10 dilution; Santa Cruz Biotechnology), anti-Ki67 (M7240, 1:200 dilution; Dako), anti-AQP1 (ab15080, 1:500 dilution; Abcom), anti-AQP1 (HPA019206, 1:1000 dilution; Sigma), anti-AQP2 (A7310, 1:200 dilution; Sigma), Biotinylated LTL (B-1325, 1:300 dilution; Vector), anti-E-Cad (AF748, 1:100 dilution; R&D), anti-UMOD (HPA043420, 1:1000 dilution; Sigma), anti-NCC (HPA028748, 1:200 dilution, Sigma), negative control mouse IgG1 (X0931, 1:5 dilution, Dako), and negative control rabbit immunoglobulin fraction (X0936, 1:3000 dilution, Dako).

**Western blotting**. For western blotting, tissues were fixed by 10% trichloroacetic acid immediately after freeze fracturing to avoid protein degradation. The fixed samples were treated with 9 M urea, 2% Triton X-100, and 1% dithiothreitol solution to solubilize protein. After sonication, 10% lithium dodecyl sulfate solution was applied to the samples. The samples were sonicated again and were separated by SDS-PAGE. After blotting, the membranes were blocked with 5% skim milk/0.1% Tween-20/TBS for 1 h at room temperature and incubated with anti-PC1 antibodies (7E12, sc-130554, 1:100 dilution, Santa Cruz Biotechnology; E8, 8C3C10, 1:3000 dilution, Baltimore PKD Core Center; 5F4D2, MABS1252, 1:200 dilution, Millipore) in 1% skim milk/0.1% Tween-20/TBS overnight at 4 °C. After washing three times with 0.1% Tween-20/TBS, the membranes were incubated with HRP-conjugated secondary antibodies in 1% skim milk/0.1% Tween-20/TBS for 1 h at room temperature. After washing three times with 0.1% Tween-20/TBS, immunoreactive proteins were detected with enhanced chemiluminescence (Cemi-Lumi One Super, Nacalai) and an ImageQuant LAS 4000 imager (GE Healthcare). After stripping with WB Stripping Solution Strong (Nacalai), beta-actin proteins were detected with Anti-beta-actin pAb-HRP-DirectT (PM053-7, MBL).

**Construction of the gRNA/Cas9 protein RNP complex**. The crRNA, tracrRNA, and HiFi Cas9 proteins were purchased from Integrated DNA Technologies. The construction of the gRNA/Cas9 protein RNP complex was performed according to the manufacturer's instructions.

**Construction of the knock-in vector and long ssODN**. To construct the floxed exon 4 knock-in vector, three amplified PCR products containing loxP sequences were cloned into the XhoI–NotI site of pBRBlue II, then two amplified PCR products containing two substituted bases from the vector were cloned into the XhoI–NotI site of pBRBlue II. An amplified PCR product from the vector was used as the template for the long ssODN, which was produced using the Guide-it Long ssDNA Production System (TaKaRa). The primers for PCR are listed in Supplementary Table 2.

**Co-injection of RNP complex and sperm and electroporation**. MII oocytes were collected as described above. The RNP complex was co-injected with sperm during ICSI[38]. MII oocytes and sperm were washed in a drop of 200 ng/μl RNP complex in opti-MEM (Invitrogen), and ICSI was performed in the drop. Electroporation of the RNP complex into ICSI embryos was performed using a Super Electroporator NEPA21 (Nepa Gene). In brief, 10 min after ICSI, zygotes were placed in the glass chamber between 1-mm-gap platinum electrodes (Nepa Gene); this chamber was filled with 6 μl of opti-MEM containing 200 ng/μl RNP complex[39]. The poring pulse parameters were as follows: voltage, 40 V; pulse width, 3.5 ms; pulse interval, 50 ms; and number of pulses, +4. The transfer pulse parameters were as follows: voltage, 5 V; pulse width, 50 ms; pulse interval, 50 ms; and number of pulses, ±5.

**Blood analysis**. In the hematological analysis, we examined leukocyte count (WBC), erythrocyte count (RBC), hemoglobin concentration (HGB), hematocrit value (HCT), mean corpuscular volume (MCV), mean corpuscular hemoglobin (MCH), mean corpuscular hemoglobin concentration (MCHC), and platelet count (PLT) using ADVIA120 (Siemens Healthcare Diagnostics Manufacturing). In the blood biochemical test, we examined aspartate transaminase (AST), alanine transaminase (ALT), lactate dehydrogenase (LDH), total bilirubin (T-bil), total protein (TP), albumin (ALB), globulin (Glob), albumin/globulin ratio (A/G), glucose (GLU), urea nitrogen (UN), inorganic phosphorus (IP), calcium (Ca), sodium (Na), potassium (K), chloride (Cl), beta2-microglobulin (B2m), triglycerides (TG), total cholesterol (T-cho), creatinine (Cre), and uric acid (UA) with JCA-BM6070 (JEOL), and cystatin C (Cys C) with a human cystatin ELISA (BioVender—Laboratorni Medicina).

**Statistical analysis**. One-way ANOVA and post hoc contrasts were used for the comparisons in Figs. 1c, 4c, e, h, 7e, and Supplementary Figs. 2c, 5b, and 6f.

**Reporting summary**. Further information on research design is available in the Nature Research Reporting Summary linked to this article.

## Data availability

The source data underlying Figs. 1c, 4c, e, h, 5b, 7e and 8b and Supplementary Figs 1c, 1d, 1e, 2c, 4d, 5c and 6f and Supplementary Table 1 are provided as a Source Data file.

The datasets generated or analyzed during the current study are available from the corresponding author on reasonable request.

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

## Acknowledgements

We are grateful to Dr. Mitinori Saitou (Kyoto University), Dr. Hitoshi Niwa (Kumamoto University), and Dr. Satoru Takahashi (Tsukuba University) for invaluable comments and advice on this work. We thank the following people for kindly providing materials for this study: Dr. Hitoshi Niwa for the pBRBlueII vector, Dr. Masahito Ikawa (Osaka University) for the pCAG-EGxxFP vector, and Dr. Qian Feng and Dr. Terry Watnick (University of Maryland, Baltimore Polycystic Kidney Disease Research and Clinical Core Center) for an anti-PC1 antibody (E8) and a PC1 expression vector. Studies utilized resources provided by the NIDDK-sponsored Baltimore Polycystic Kidney Disease Research and Clinical Core Center, P30 DK090868. This study was supported in part by grants from PRESTO, funded by the Japan Science and Technology Agency (JST), JSPS Grant-in-Aid for challenging Exploratory Research Grant Number 19K22363 to M.E.; by JSPS KAKENHI Grant Number JP17K08135 to T.T.; by a grant from the World Premier International Research Center Initiative (WPI) to M.E., M.N., and T.T.; and by an in-house grant from Shiga University of Medical Science to M.E. and T.T.

## Author contributions

T.T. conceived, designed, and performed the experiments, analyzed and interpreted the data, and wrote the paper. Kenichi Kobayashi, M.N., and Kahoru Kitajima performed the experiments. C.I., Y.S., H.T., and J.M. generated the monkeys. I.K., T.N., K.F., T.I., H.I., I.I., and Shinichiro Nakamura conducted phenotypic analysis of live monkeys. S.K., H.M., and A.K. interpreted the data. R.N. and Saori Nishio interpreted the data and edited the paper. M.E. conceived, interpreted the data, wrote the paper, and supervised the overall project.

## Competing interests

The authors declare no competing interests.
