## [Peer Review File · Nature Communications]

Reviewers' comments:

Reviewer #1 (Remarks to the Author):

The authors report progress toward developing an ADPKD model in cynomolgus monkeys. They have used CRISPR/Cas9 to introduce PKD1 mutations. Despite large amount of work, it is only work in progress. The affected animals with a severe phenotype are biallelically affected mosaics. The authors have also generated PKD1 heterozygote animals using allele-specific targeting, but their phenotype at a young age is very mild. Since the life span of cynomolgus monkeys is 25-30 years and they do not reach sexual maturity until they are 4-5 years of age, longer follow-up will be interesting. However, the questionable practicality and superiority of this model for preclinical testing, compared to other large animal models, raise the issue of whether use of primates for this purpose is justified. The authors should address this in the Discussion.

Reviewer #2 (Remarks to the Author):

The manuscript by Tsukiyama and colleagues described the generation of a monkey model of polycystic kidney disease by CRISPR/Cas9 gene editing technology. The report highlighted the commitment of the team in creating the novel monkey model that successfully captured the key pathological changes and clinical blood biochemistry observed in human patients. While the gene editing approach itself is not novel, the high efficiency in creating gene targeted monkey is impressive.

The success of the model was primarily based on the evidences of kidney pathologies in monkeys at various stages of pregnancies and ages which suggest that a high heterogeneity of gene editing events were occurred and was supported by the high mosaic rates in embryos and monkeys. To minimize the non-specific targeting event, the team modified the manipulation approach and developed an allele specific targeting approach based on the polymorphism between the maternal and paternal allele of the PKD1-coding region. By modify the injection sequence and the use of gRNA/Cas9 protein RNP complex; they reported high successful rate in creating non-mosaic gene targeted monkeys.

In addition to kidney pathologies, renal functions of at six and 12 months of age was determined by blood biochemistry which showed no remarkable abnormalities in monkey with cyst formation and is consistent in clinical study as authors indicated.

Overall, based on clinical and pathological evidences, the team has successfully generated a monkey model that replicates human autosomal dominant polycystic kidney disease which could be a useful model to advance the field of ADPKD. The success in generating non-mosaic monkey was also impressive but it is unclear how mosaicism was defined. For example, how many tissues were analyzed and if PKD1 mutations can be transmitted through the germline?

Other concerns:

1. What was the tissue used for genotyping of the deceased monkeys? How many tissues were used for genotyping and if all tissues share identical or different mutations that constitute mosaicism. Is mosaic was defined as multiple mutations were identified in kidney only?
2. The authors indicated that placental abruption is common in cynomolgus monkey but it was 40% higher in PKD1 mutated fetuses that seem significant and deserve an in-depth discussion such as possible consequence of off-target effect and the extent of gene editing events in the resulted fetuses and miscarriage monkeys.
3. One of the major concerns in gene editing animal research is high mosaic rate and off-target effect. While the manuscript has described mosaic animals but it is unclear a throughout assessment in available tissues from deceased animals was performed or primarily based on kidney tissue as indicated. The team falls short to take the advantage of the deceased animals to fully assess the extent of mosaicism in gene targeted monkeys. In addition to the concerns on mosaicism, there is no discussion on off-target analysis especially for the high rate of miscarried fetuses. Even placental abruption is common in cynomolgus monkey; ~40% higher in PKD1 mutated fetuses is significant that deserve a closer look at the potential issues of the approach.
4. Although the team has claimed the generation of non-mosaic animal with modified approach, it is unclear how many tissues have been analyzed to determine the non-mosaic status or primarily focused on kidney tissues. It is also unclear about the age of the animals and whether germline transmission can soon be determined.

5. At the end, the team presented data on the generation of floxed allele in monkey embryos which does not seem to be necessary and relatively premature to present.

Reviewer #3 (Remarks to the Author):

The manuscript by Tsukiyama et al., describes generation of monkey models for polycystic kidney disease using the CRISPR-Cas9 technology, followed by the thorough characterization of the models demonstrating their utility for PKD disease research. The experiments are, logical, systematic and are nearly complete. Except for a few concerns, I recommend this manuscript for publication.

Comments/suggestions:

Line 126-142: The authors describe genotype and phenotype of the kidney samples (histological classification) collected from the live born/aborted monkeys. However, it was hard to understand (from the text) if there was any correlation of genotype and phenotype (especially heterozygotes and homozygotes, if any).

The table in figure 1b is somewhat confusing and not easy for reader to follow it quickly. My understanding is that some embryos were analyzed at blastocyst stage (labelled as “analyzed” column) and some were used for embryo transfer (“ET” column). This is not described clearly in the text/legend. In addition to adding such details (in the text and legend), moving the ET column to the end (as the last column) would be better (to avoid confusion to the readers). Did the authors not analyze or ET all of the surviving embryos? If so, authors could describe this to avoid readers getting confused about what happened to the remaining surviving embryos.

Adding some more discussion about PKD disease progression in humans (in comparison to the monkey model) particularly to emphasize species differences (such as disease onset and progression) would be helpful. This is to support author’s statement in line numbers 257-261 (which reads as follows: Given that monkeys have longer life spans than rodents, and the cyst formation in our primate models resembled not only that in late-stage human patients but also that in pediatric patients, we can monitor the disease state over the long term and thereby elucidate the molecular mechanisms of ADPKD and assessing drug efficacy and toxicity). This comment is also relevant to a sentence in line numbers 372-374 (We generated PKD1 heterozygote nonhuman primates that formed cysts in a similar way as human pediatric patients).

Line 354-356: Authors state that “Introduction of Cre recombinase into this floxed allele would further accelerate the study of ADPKD in primates”. As I understand, the authors created floxed alleles only in embryos and those embryos were harvested for analysis. It is not clear how Cre can be introduced to embryos after harvesting them (?) I am not sure introducing Cre to such embryos (to study gene deletion effect) would be that easy. It would be helpful if they can provide some description on how Cre can be introduced to such embryos for studying ADPKD. On the other hand if there are no methods available to do this, they can remove/revise this sentence appropriately.

Fig 8b. The RFLP genotyping strategy is not clear from the figure. Location of genotyping primers in the schematic would be helpful. Also, it was not clear how the authors differentiated cis vs trans insertions of LoxP sites or partial insertions of LoxPs (if they indeed occurred in their experiments). Even though this is a minor point (which may not change the overall conclusions of the paper), thorough analysis of correct targeting efficiency would be necessary especially for floxed alleles. If samples are still available it would be suggested to do analysis of samples as described in Miura et al Nature Protocols volume 13, pages 195–215 (2018), and discuss those results.

The authors used PCR templates for generating long ssDNAs (not the nicking method). However, the reference 40 (that the authors cited for this method) describes nicking method and also this reference does not describe floxed allele generation. Therefore, in addition to reference 40, the authors could cite other references (for example, Miyasaka et al., BMC Genomics volume 19, Article number: 318 (2018) and Quadros et al, Genome Biology volume 18, Article number: 92 (2017) which actually describe nicking method and/or floxed allele generation methods).

Discussion section is somewhat short. It can be expanded further by adding discussion about a few more points (some suggestions of points would be: to discuss on genotype-phenotype correlation; authors observation and/or interpretation of interhomolog repair mechanism from the experiment in figure 7 (if their observation was similar to, or opposite to, that of Ma et al interpretation)

A couple typos:

Line 839: ssDNS => ssDNA

Legend of Supp figure 1 “294FT=>293FT”

Thank you very much for your helpful suggestions regarding our manuscript. During the revision process we made necessary changes according to the reviewers' suggestions. Our responses to the comments are below.

Reviewers' Comments:

Reviewer #1:

The authors report progress toward developing an ADPKD model in cynomolgus monkeys. They have used CRISPR/Cas9 to introduce PKD1 mutations. Despite large amount of work, it is only work in progress. The affected animals with a severe phenotype are biallelically affected mosaics. The authors have also generated PKD1 heterozygote animals using allele-specific targeting, but their phenotype at a young age is very mild. Since the life span of cynomolgus monkeys is 25-30 years and they do not reach sexual maturity until they are 4-5 years of age, longer follow-up will be interesting. However, the questionable practicality and superiority of this model for preclinical testing, compared to other large animal models, raise the issue of whether use of primates for this purpose is justified. The authors should address this in the Discussion.

We are very grateful to you for bringing these issues to our attention. Other large animal models, such as those involving induced mutations in pigs and spontaneous mutations in cats and dogs, are also useful for preclinical testing. However, monkeys are more similar to humans genetically and physiologically than these animals. Additionally, like in humans, genetic diversity is higher in monkeys than in other model animals. These features are advantageous for preclinical studies. Certainly, it is true that the use of monkeys would involve expensive and long-term studies. However, although ADPKD is a very common monogenic disease that occurs very frequently, it is also true that treatment methods are still limited despite several decades of research thus far. Even if it takes time and money, we believe it is time to try a new research approach. In contrast to mice, which require conditional knockout to mimic the human phenotype, it is feasible to investigate the process of cyst formation in a more natural state using our monkey model. Since heterozygotes without mosaicism can be prepared, we believe that they can be studied genetically in a relatively homogeneous population. Although this will of course take time, in the future we can perform this research under more genetically homogeneous conditions in family pedigrees by analyzing the F1 generation. We have

not yet evaluated the fertility of the monkeys we generated, but given that human ADPKD patients are fertile, we believe that these monkeys will be fertile and can generate multiple pedigrees. At our facility we can currently house about 800 monkeys, and we plan to increase the capacity in the future. We also plan to provide these monkeys to other ADPKD researchers. The pathology in heterozygotes was similar to that in humans, so the same pathological progression can be expected in the F1 generation. Therefore, we hope that elucidating the mechanisms by which cysts develop and worsen may identify treatment that can begin in childhood. We added this information on P. 12, L. 25–P. 13, L. 17.

Reviewer #2:

The manuscript by Tsukiyama and colleagues described the generation of a monkey model of polycystic kidney disease by CRISPR/Cas9 gene editing technology. The report highlighted the commitment of the team in creating the novel monkey model that successfully captured the key pathological changes and clinical blood biochemistry observed in human patients. While the gene editing approach itself is not novel, the high efficiency in creating gene targeted monkey is impressive.

The success of the model was primarily based on the evidences of kidney pathologies in monkeys at various stages of pregnancies and ages which suggest that a high heterogeneity of gene editing events were occurred and was supported by the high mosaic rates in embryos and monkeys. To minimize the non-specific targeting event, the team modified the manipulation approach and developed an allele specific targeting approach based on the polymorphism between the maternal and paternal allele of the PKD1-coding region. By modify the injection sequence and the use of gRNA/Cas9 protein RNP complex; they reported high successful rate in creating non-mosaic gene targeted monkeys.

In addition to kidney pathologies, renal functions of at six and 12 months of age was determined by blood biochemistry which showed no remarkable abnormalities in monkey

with cyst formation and is consistent in clinical study as authors indicated.

Overall, based on clinical and pathological evidences, the team has successfully generated a monkey model that replicates human autosomal dominant polycystic kidney disease which could be a useful model to advance the field of ADPKD. The success in generating non-mosaic monkey was also impressive but it is unclear how mosaicism was defined. For example, how many tissues were analyzed and if PKD1 mutations can be transmitted through the germline?

We are grateful for your comments and useful suggestions that have helped us improve our paper. The items that need to be addressed in detail are explained in the point-by-point responses below.

Other concerns:

1. What was the tissue used for genotyping of the deceased monkeys? How many tissues were used for genotyping and if all tissues share identical or different mutations that constitute mosaicism. Is mosaic was defined as multiple mutations were identified in kidney only?

In some monkeys (Mild #1, Severe #1, Severe #3), many tissues (kidney, liver, spleen, stomach, pancreas, small intestine, colon, uterus, bladder, ovary, adrenal glands, thymus, thyroid, heart, lung, brain, cerebellum, skin, placenta, amnion, and umbilical cord) were used for genotyping. However, we could not detect clear variations in genotypes among tissues. Given that mutations were induced early in development, it was appropriate to observe mosaicism within individual tissues, not between tissues. Therefore, in other monkeys, genomic DNA samples were collected from the kidneys of dead animals, and from the placentas, amnions, and umbilical cords of living monkeys to reduce invasiveness. We added this information on P. 4, L. 33–P. 5, L. 8.

2. *The authors indicated that placental abruption is common in cynomolgus monkey but it was 40% higher in PKD1 mutated fetuses that seem significant and deserve an in-depth discussion such as possible consequence of off-target effect and the extent of gene editing events in the resulted fetuses and miscarriage monkeys.*

We are very grateful to you for bringing these issues to our attention. It is difficult to clearly demonstrate off-target sequence alterations in a genome-wide manner due to limited genomic information in monkeys. Therefore, we used two different gRNAs to show that observed phenotypes were caused by on-target mutations. Interestingly, compared with WT monkeys, we observed higher abortion rates not only with exon 2 targeting but also with exon 4 targeting. In mice, it was shown that loss of *Pkd1* caused placental defects (Garcia-Gonzalez et al., PLoS One, 2010). Thus, we suspect that abnormal function of *PKD1* may be a cause of the high rate of abortion. Because that hypothesis differs from the focus of this paper, we plan to explore it in a future work. We added this information on P. 14, L. 10–16.

3. *One of the major concerns in gene editing animal research is high mosaic rate and off-target effect. While the manuscript has described mosaic animals but it is unclear a throughout assessment in available tissues from deceased animals was performed or primarily based on kidney tissue as indicated. The team falls short to take the advantage of the deceased animals to fully assess the extent of mosaicism in gene targeted monkeys. In addition to the concerns on mosaicism, there is no discussion on off-target analysis especially for the high rate of miscarried fetuses. Even placental abruption is common in cynomolgus monkey; ~40% higher in PKD1 mutated fetuses is significant that deserve a closer look at the potential issues of the approach.*

As we mentioned above, many tissues from several monkeys were used for genotyping. However, we could not detect clear variations in genotypes among tissues. Given that mutations were induced early in development, it is appropriate to observe mosaicism within individual tissues, not between tissues. Therefore, in other monkeys, genomic DNA samples were collected from the kidneys of dead animals, and from the placentas, amnions, and umbilical cords of living monkeys to reduce invasiveness. We

added this information on P. 4, L. 33–P. 5, L. 8.

4. *Although the team has claimed the generation of non-mosaic animal with modified approach, it is unclear how many tissues have been analyzed to determine the non-mosaic status or primarily focused on kidney tissues. It is also unclear about the age of the animals and whether germline transmission can soon be determined.*

In living monkeys, genomic DNA samples were collected from placentas, amnions, and umbilical cords. In dead monkeys, kidneys or aborted fetal tissues were used. All living monkeys are under 3 years old, so we have not evaluated the fertility of the monkeys we generated, but given that human ADPKD patients are fertile, we believe that these monkeys will also be fertile and can generate multiple pedigrees. We added this discussion on P. 13, L. 12–14.

5. *At the end, the team presented data on the generation of floxed allele in monkey embryos which does not seem to be necessary and relatively premature to present.*

Knock-in of large sequences into early NHP embryos is still challenging, and no studies thus far have succeeded in generating floxed alleles in NHPs. Therefore, we believe that this information deserves to be reported.

Reviewer #3:

The manuscript by Tsukiyama et al., describes generation of monkey models for polycystic kidney disease using the CRISPR-Cas9 technology, followed by the thorough characterization of the models demonstrating their utility for PKD disease research. The experiments are, logical, systematic and are nearly complete. Except for a few concerns, I recommend this manuscript for publication.

Thank you very much for your positive evaluations and insightful comments on our manuscript. The items that need to be addressed in detail are explained in the point-by-point responses below.

Comments/suggestions:

Line 126-142: The authors describe genotype and phenotype of the kidney samples (histological classification) collected from the live born/aborted monkeys. However, it was hard to understand (from the text) if there was any correlation of genotype and phenotype (especially heterozygotes and homozygotes, if any).

Thank you very much for your helpful suggestions regarding our manuscript. Although the phenotype is variable in exon 2 targeting, due to mosaicism, the severity of the phenotype tended to correlate with the mutation rate. We clarified this on P. 5, L. 21–23.

The table in figure 1b is somewhat confusing and not easy for reader to follow it quickly. My understanding is that some embryos were analyzed at blastocyst stage (labelled as “analyzed” column) and some were used for embryo transfer (“ET” column). This is not described clearly in the text/legend. In addition to adding such details (in the text and legend), moving the ET column to the end (as the last column) would be better (to avoid confusion to the readers). Did the authors not analyze or ET all of the surviving embryos? If so, authors could describe this to avoid readers getting confused about what happened to the remaining surviving embryos.

As you mentioned, some embryos were analyzed in the blastocyst stage and some were used for embryo transfer. We used blastocysts for genotyping analysis (Analyzed) or embryo transfer (ET), while the remaining embryos were frozen. Overall, 207 of 423 embryos that survived after injection developed into blastocysts. Among the 207 embryos, 71 were used for genotyping, 86 were used for embryo transfer to generate monkeys, and the remaining 50 were frozen. We added this information on P. 3, L. 31–33

and moved the ET column to the end of the table in Figure 1b.

Adding some more discussion about PKD disease progression in humans (in comparison to the monkey model) particularly to emphasize species differences (such as disease onset and progression) would be helpful. This is to support author's statement in line numbers 257-261 (which reads as follows: Given that monkeys have longer life spans than rodents, and the cyst formation in our primate models resembled not only that in late-stage human patients but also that in pediatric patients, we can monitor the disease state over the long term and thereby elucidate the molecular mechanisms of ADPKD and assessing drug efficacy and toxicity). This comment is also relevant to a sentence in line numbers 372-374 (We generated PKD1 heterozygote nonhuman primates that formed cysts in a similar way as human pediatric patients.

Generally, active therapeutic interventions in pediatric patients, including molecular diagnosis, are not recommended, and research in children of ADPKD patients is ethically difficult because ADPKD shows 100% penetrance and there is no definitive therapy. In humans, therefore, it has only been revealed that cysts have developed since childhood, and the details are unknown. Additionally, since rodent models do not precisely recapitulate the disease state in humans, the pathological process from the presymptomatic stage to the early stage in humans remains a black box. Here, we identified the lineage identities of cyst epithelia, representing the earliest manifestations of the disease. This study shows for the first time the initial dynamics of a disease in which a small cyst develops from a distal tubule. Thus, although the collecting ducts are the only drug target in adulthood patients, our results suggest that the distal tubules may be a novel drug target in pediatric patients for whom there is currently no clinical intervention. We added this information on P. 13, L. 18–30.

Line 354-356: Authors state that "Introduction of Cre recombinase into this floxed allele would further accelerate the study of ADPKD in primates". As I understand, the authors created floxed alleles only in embryos and those embryos were harvested for analysis. It is not clear how Cre can be introduced to embryos after harvesting them (?) I am not sure

introducing Cre to such embryos (to study gene deletion effect) would be that easy. It would be helpful if they can provide some description on how Cre can be introduced to such embryos for studying ADPKD. On the other hand if there are no methods available to do this, they can remove/revise this sentence appropriately.

For Cre delivery, we plan to use an adenovirus to generate renal tubule-specific PKD1 KO after the generation of monkeys with floxed alleles. We clarified this on P. 12, L. 12–14.

Fig 8b. The RFLP genotyping strategy is not clear from the figure. Location of genotyping primers in the schematic would be helpful. Also, it was not clear how the authors differentiated cis vs trans insertions of LoxP sites or partial insertions of LoxPs (if they indeed occurred in their experiments). Even though this is a minor point (which may not change the overall conclusions of the paper), thorough analysis of correct targeting efficiency would be necessary especially for floxed alleles. If samples are still available it would be suggested to do analysis of samples as described in Miura et al Nature Protocols volume 13, pages 195–215 (2018), and discuss those results.

Genomic fragments containing the outer regions of homologous arms were amplified by PCR (Fig. 8a, blue triangles), and the fragments were cloned into vectors and sequenced. The sequencing analysis confirmed *cis* insertion of LoxP sites in all 3 floxed embryos. In one embryo (Experiment 1, #4), we detected a single LoxP insertion. Additionally, as in the methods described by Miura et al (Nature Protocols), we confirmed the sequence fidelity of junctional regions in two of three floxed embryos. To clarify this, we added the locations of genotyping primers in Figure 8a and added further information on P. 12, L. 5–10.

The authors used PCR templates for generating long ssDNAs (not the nicking method). However, the reference 40 (that the authors cited for this method) describes nicking method and also this reference does not describe floxed allele generation. Therefore, in

addition to reference 40, the authors could cite other references (for example, Miyasaka et al., BMC Genomics volume 19, Article number: 318 (2018) and Quadros et al, Genome Biology volume 18, Article number: 92 (2017) which actually describe nicking method and/or floxed allele generation methods).

We added these citations. Thank you very much.

Discussion section is somewhat short. It can be expanded further by adding discussion about a few more points (some suggestions of points would be: to discuss on genotype-phenotype correlation; authors observation and/or interpretation of interhomolog repair mechanism from the experiment in figure 7 (if their observation was similar to, or opposite to, that of Ma et al interpretation)

We added a description of genotype–phenotype correlation in the main text.

We sequenced only positive samples in the RFLP assay, and were unable to obtain clear evidence for interhomolog repair because in these samples we could detect mutated or floxed sequences but not the WT sequence. We think that further studies are required to determine the incidence rate of interhomolog repair.

Therefore instead of that information, we added material in the discussion section about the practicality and superiority of this model for preclinical testing, compared to other animal models.

A couple typos:

Line 839: ssDNS => ssDNA

Legend of Supp figure 1 “294FT=>293FT”

Thank you very much. We corrected the descriptions.

REVIEWERS' COMMENTS:

Reviewer #1 (Remarks to the Author):

The authors have appropriately addressed the issues raised in the previous review.

Reviewer #2 (Remarks to the Author):

The authors have addressed and clarified my concerns on mosaicism and tissues assessment with fair assessment. Although reviewer still consider the Knock-in embryo data might not necessary for this specific manuscript and provide additional information on the model itself, will leave the decision to the editorial and authors.

Reviewer #3 (Remarks to the Author):

The authors have addressed the comments satisfactorily. One minor change: regarding the sentence "For Cre delivery, we plan to use an adenovirus to generate renal tubule-specific PKD1 KO after the generation of monkeys with floxed alleles", the authors can modify it as "Once the floxed monkeys are generated, deletion of PKD1 in renal tubule cells can be achieved by viral delivery of Cre enzyme.",

Thank you very much for your helpful suggestions regarding our manuscript. During the revision process we made necessary changes according to the reviewers' suggestions. Our responses to the comments are below.

Reviewers' Comments:

Reviewer #1 (Remarks to the Author):

The authors have appropriately addressed the issues raised in the previous review.

Thank you very much for your helpful suggestions regarding our manuscript.

Reviewer #2 (Remarks to the Author):

The authors have addressed and clarified my concerns on mosaicism and tissues assessment with fair assessment. Although reviewer still consider the Knock-in embryo data might not necessary for this specific manuscript and provide additional information on the model itself, will leave the decision to the editorial and authors.

We are grateful for your comments and useful suggestions that have helped us improve our paper.

Reviewer #3 (Remarks to the Author):

The authors have addressed the comments satisfactorily. One minor change: regarding the sentence "For Cre delivery, we plan to use an adenovirus to generate renal tubule-specific PKD1 KO after the generation of monkeys with floxed alleles", the authors can modify it as "Once the floxed monkeys are generated, deletion of PKD1 in renal tubule cells can be achieved by viral delivery of Cre enzyme.",

Thank you very much for your positive evaluations and insightful comments on our manuscript. We modified the sentence as you suggested.